# The rate of transient beta frequency events predicts behavior across tasks and species

Hyeyoung Shin[1]*, Robert Law[1,2], Shawn Tsutsui[1], Christopher I Moore[1], Stephanie R Jones[1,2]

[1]Department of Neuroscience, Brown University, Providence, United States; [2]Center for Neurorestoration and Neurotechnology, Providence VA Medical Center, Providence, United States

**Abstract** Beta oscillations (15-29Hz) are among the most prominent signatures of brain activity. Beta power is predictive of healthy and abnormal behaviors, including perception, attention and motor action. In non-averaged signals, beta can emerge as transient high-power 'events'. As such, functionally relevant differences in averaged power across time and trials can reflect changes in event number, power, duration, and/or frequency span. We show that functionally relevant differences in averaged beta power in primary somatosensory neocortex reflect a difference in the number of high-power beta events per trial, i.e. event rate. Further, beta events occurring close to the stimulus were more likely to impair perception. These results are consistent across detection and attention tasks in human magnetoencephalography, and in local field potentials from mice performing a detection task. These results imply that an increased propensity of beta events predicts the failure to effectively transmit information through specific neocortical representations.
DOI: https://doi.org/10.7554/eLife.29086.001

*For correspondence:
shinehyeyoung@gmail.com

Competing interests: The authors declare that no competing interests exist.

## Introduction

Beta frequency oscillations are among the most prominent signatures of brain activity. Modulations in beta power correlate with perceptual and motor demands (*Bauer et al., 2006*; *Bressler and Richter, 2015*; *Engel and Fries, 2010*; *Haegens et al., 2011*; *Jones et al., 2010*; *Miller et al., 2012*; *Neuper and Pfurtscheller, 2001*; *Sacchet et al., 2015*; *Siegel et al., 2008*; *van Ede et al., 2011*) and abnormalities in beta are a biomarker for neuropathology (e.g. in Parkinson's disease, Alzheimer's disease and Autism Spectrum Disorder [*Karageorgiou and Vossel, 2017*; *Khan et al., 2015*; *Little and Brown, 2014*]). Yet, how and why beta impacts function is debated. The temporal signatures of rhythmic activity may prove crucial to understanding their importance in brain function (*Cole and Voytek, 2017*; *Jensen et al., 2016*; *Jones, 2016*). Yet, most studies reporting functional correlates of brain rhythms rely on averaging across time and/or trials, where the temporal dynamics of rhythmic activity is lost.

Several recent studies have appreciated that, while some rhythms are sustained for several cycles (e.g. occipital alpha rhythms during eye closure, slow wave sleep rhythms), rhythmic activity can often be transient in non-averaged data (*Feingold et al., 2015*; *Jones, 2016*; *Jones et al., 2009*; *Lundqvist et al., 2016*; *Sherman et al., 2016*; *Tinkhauser et al., 2017*). In such studies, high power beta frequency activity (15–29 Hz) emerges as brief events typically lasting <150 ms. Transient beta has been reported in many brain areas including somatosensory (*Jones et al., 2009*; *Sherman et al., 2016*; *Ziegler et al., 2010*), motor (*Feingold et al., 2015*; *Rule et al., 2017*), occipital (*Freyer et al., 2009*), and frontal neocortex (*Lundqvist et al., 2016*; *Sherman et al., 2016*), and basal ganglia structures (*Bartolo and Merchant, 2015*; *Feingold et al., 2015*; *Tinkhauser et al., 2017*), and

across recording modalities and species (*Sherman et al., 2016*). Further, differences in the accumulated density of transient beta bursts across trials predicts functionally relevant differences in averaged beta power reflecting motor and cognitive demands in monkey local field potentials (LFP) data (*Feingold et al., 2015*; *Lundqvist et al., 2016*).

We have previously shown, using magnetoencephalography (MEG), that lower prestimulus beta power (averaged in the 1 s prestimulus time window) in the human primary somatosensory neocortex (SI) predicts an increase in the probability of detection of tactile stimuli at perceptual threshold (*Jones et al., 2010*). Correspondingly, shifts in spatial attention modulate post-cue/prestimulus beta power, such that average power decreases in the attended SI location (*Jones et al., 2010*; *Sacchet et al., 2015*). These prior studies did not investigate the relationship between the transient nature of the beta events on specific trials and function.

Here, we find that functionally relevant differences in averaged prestimulus beta power in human SI emerge from a difference in the number of high-power beta events per trial (e.g. the rate of beta events). This result is conserved in both detection and attention tasks (*Jones et al., 2010*; *Jones et al., 2007*): A lower rate of prestimulus beta events predicts enhanced detection of stimuli at perceptual threshold, and the rate of beta events in the attended region of the brain decreases. Moreover, non-detected trials were more likely to have a high-power beta event within approximately 200 ms of the stimulus. Importantly, these beta events were not a byproduct a sustained oscillation whose power gets dynamically modulated; rather, we find evidence that beta events may be generated by a transient, 'bursty' mechanism. These results show that differences in the rate and timing of 'bursty' beta events underlie the correlation between beta power and behavior in humans. Results were remarkably consistent in LFP recordings from awake behaving mice performing a similar tactile detection task. The strong parallel in the character of beta events across behavior, recording scales, and species indicates a fundamental consistency in the nature of this much-studied dynamic in the mammalian neocortex.

## Results

### Mean prestimulus beta power is higher on non-detected and attend-out trials

We have previously shown that prestimulus beta rhythms source localized to the hand representation in human SI are predictive of tactile perception. Beta (15–29 Hz) power averaged 1 s prestimulus shows a negative relationship to the probability of detecting a brief perceptual threshold-level (50% detection) tap to the contralateral middle finger tip (*Jones et al., 2010*). *Figure 1Aii* illustrates this result, showing that the rate of detection (hit rate, measured as percent change from mean [PCM]) decreases as averaged prestimulus beta power increases. Further, beta power plotted as a function of time was higher on non-detected trials throughout the prestimulus window (*Figure 1Ai*). Human detection data were obtained from an MEG dipole source localized to the contralateral hand area of SI in the post-central gyrus in all subjects (N = 10 subjects), see *Figure 1—figure supplement 1A* for the evoked response and further details in the Materials and methods.

Beta power recorded as LFP from SI of mice performing a similar vibrissae task (*Figure 1Bi–ii*) also shows a negative relationship to detection. These data were obtained from the SI 'barrel' neocortex contralateral to the stimulated vibrissae (N = 10 sessions from two mice), see *Figure 1—figure supplement 1A* for evoked response and further details in the Materials and methods.

These detection results imply that beta expression in a specific somatotopic representation suppresses the throughput of information in that region. If so, allocation of this dynamic to"attend-out' representations, those where distracting inputs might decrease performance, could be adaptive. Consistent with this prediction, when human subjects are cued to attend to perceptual-threshold level stimulation in the hand or foot (*Jones et al., 2010*; *Sacchet et al., 2015*), a similar negative relationship exists between prestimulus beta power and the allocation of attention (attend-in rate, measured as PCM of attend hand trial) (*Figure 1Ci–ii*; analysis was limited to detected trials to compare the effect of attention independent of detection). Human attention data were also obtained from a MEG dipole source localized to the contralateral hand area of SI (N = 10 subjects, see further details in Materials and methods and *Jones et al., 2010*).

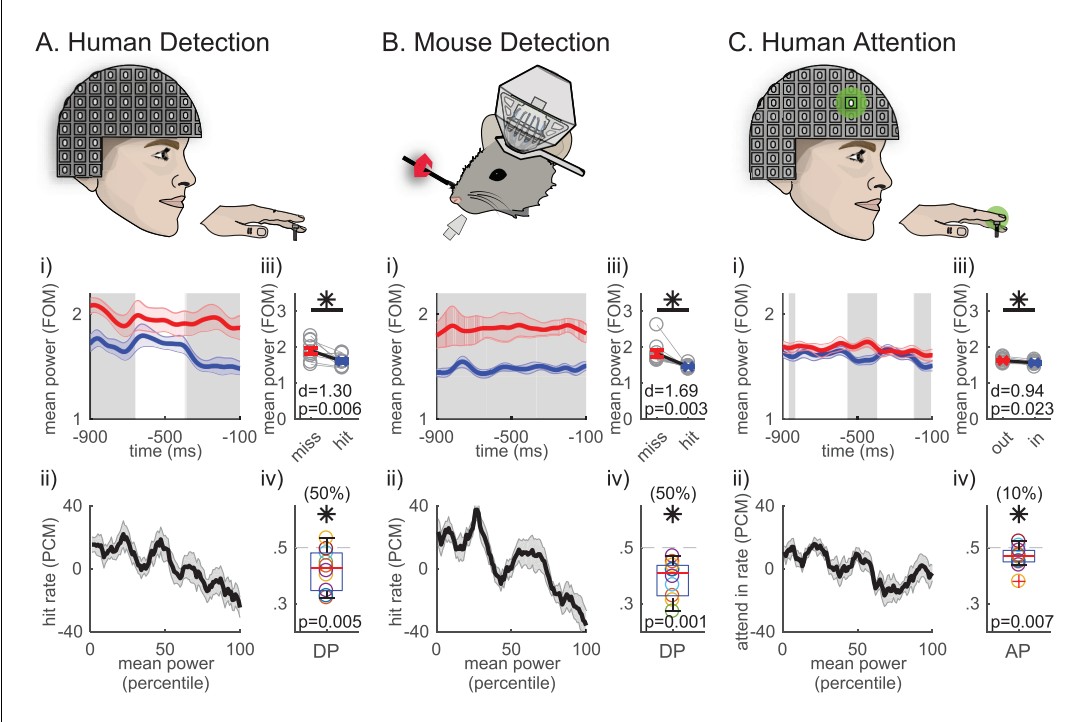

**Figure 1.** Average 1 s prestimulus beta (15–29 Hz) power is higher on non-detected/attend out trials in SI. (i) Mean prestimulus beta power as a function of time relative to stimulus onset in (**A**) human MEG during a tactile detection task (source localized data from hand area of SI; N = 10 subjects, 100 detected and non-detected trials per subject), (**B**) mouse LFP during a vibrissae deflection detection task (barrel cortex; N = 10 sessions, a mean of 120 (range 106 ~ 160) detected and non-detected trials per session, in each session the number and stimulus amplitude for detected and non-detected trials were matched), and (**C**) human MEG during a cued spatial attention task (source localized data from the hand area of SI; N = 10 subjects, 100 detected and non-detected trials per subject). 7-cycle Morlet wavelet was convolved across the 1 s prestimulus window on each trial and shown only for −900~−100 ms of the 1 s prestimulus period to circumvent regions affected by edge effects. Mean ±standard error of the mean (SEM) error bar across subjects/sessions for non-detected/attend out (red) and detected/attend in (blue) trials. Grey shaded regions show significance across subjects/sessions at p<0.05 (pointwise right-tailed Wilcoxon signed rank test). (ii) Trials were sorted from low to high trial mean prestimulus beta power (21-trial bins sliding in 1-trial steps). For each bin, hit rate and attend-in rate were calculated as percent change from mean (PCM) (mean ±SEM across subjects/ sessions). (iii) Trial mean prestimulus beta power pooled across each behavioral condition (units are factor of median, FOM; grey open circles: individual subject/session means; red and blue x: grand mean ±SEM across subjects/sessions). Asterisks denote p<0.05 right-tailed paired t-test, and d = Cohen's d effect size. (iv) Box and whisker plots of distribution of detect probability (DP)/attend probability (AP) across subjects/sessions (colored ○: each subject/session). The numbers in parentheses at the top denote the percentage of subjects/sessions with DP/AP significantly less than 0.5 (as determined by 95% confidence interval, calculated from bootstrapping 1000 times). One-sample left-tailed Wilcoxon signed rank test was applied to test whether the median across subjects/sessions was significantly less than 0.5; p-value is denoted, with asterisk if p<0.05.

DOI: https://doi.org/10.7554/eLife.29086.002

The following figure supplement is available for figure 1:

**Figure supplement 1.** Evoked response to suprathreshold sensory input and beta power on catch trials.
DOI: https://doi.org/10.7554/eLife.29086.003

To determine whether the negative relationships between averaged beta power and behavior visualized in *Figure 1ii* produced a significant difference across behavioral conditions, we compared the grand averaged prestimulus power pooled across conditions in each dataset (*Figure 1iii*). All three datasets showed significance, such that the pooled average was significantly higher on non-detected/attend out trials (p<0.05, right-tailed paired t-test). Quantification of the corresponding effect size showed the difference across behavioral conditions was larger for detection than attention (see Cohen's d values in *Figure 1* panels iii).

To further investigate the potential impact of prestimulus beta power on behavior, we conducted an ideal observer analysis (*Nienborg et al., 2012*). Trial mean prestimulus beta power denotes beta power averaged across the 1 s prestimulus period, and across the 15–29 Hz beta band. The area under the receiver operating characteristic curve of the ideal observer is termed detect probability

(DP) and attend probability (AP) for detection and attention tasks, respectively. A DP or AP value of 0.5 indicates no predictability, and <0.5 indicates that higher trial mean prestimulus beta power was predictive of non-detected or attend-out trials. Box plot distributions of these values across subjects or sessions are plotted in *Figure 1iv*. In all three datasets, the median of DP and AP distributions were significantly <0.5, such that higher power indicated the subject was less likely to detect or attend to the stimulus (*Figure 1iv* asterisks p<0.05, p-values from one-sample left-tailed Wilcoxon signed rank test). While this effect was significant in all three datasets, the effect was stronger in the detection datasets, where in 50% of human subjects and mouse sessions high beta was significantly predictive of non-detection, as compared to the attention dataset where only 10% of the subjects showed significance (*Figure 1iv*, see parentheses above each plot, bootstrapped 1000 times to determine 95% confidence interval for each subject/session. DP/AP is significantly less than 0.5 if the upper boundary of the bootstrapped confidence interval is less than 0.5).

To determine if the difference in prestimulus beta power between non-detected and detected trials might be due to a lower criterion for perception and/or enhanced motor readiness, we assessed the relationship between correct reject and false alarm trials and beta power in the mouse dataset. Averaged beta power was not significantly higher in correct rejections compared to false alarms across sessions, arguing strongly against the possibility that the difference was due to modulation of the criterion or motor readiness (*Figure 1—figure supplement 1B*; right-tailed paired t-test). This analysis was not possible in the human detection data due to task design and low false alarm rates. We restricted all further analyses to trials at perceptual threshold.

## Beta emerges as brief events on individual trials

The results in *Figure 1* establish that prestimulus beta power is predictive of perception and attentional allocation. Such averaging of power across time, frequency and trials is typical for the investigation of the relationship between brain rhythms and function. A key step in understanding how beta impacts function is to identify features in the non-averaged signal that contribute to this relationship.

Beta emerges as a transient surge in power (i.e. events) on individual trials in our data (*Figure 2ii*). Because power spectral values are non-negative, the accumulation of events across trials creates a continuous band of activity in the average, often misinterpreted as a sustained rhythm (*Figure 2i*) (*Jones, 2016*). We find that this observation holds across tasks and species (*Figure 2*).

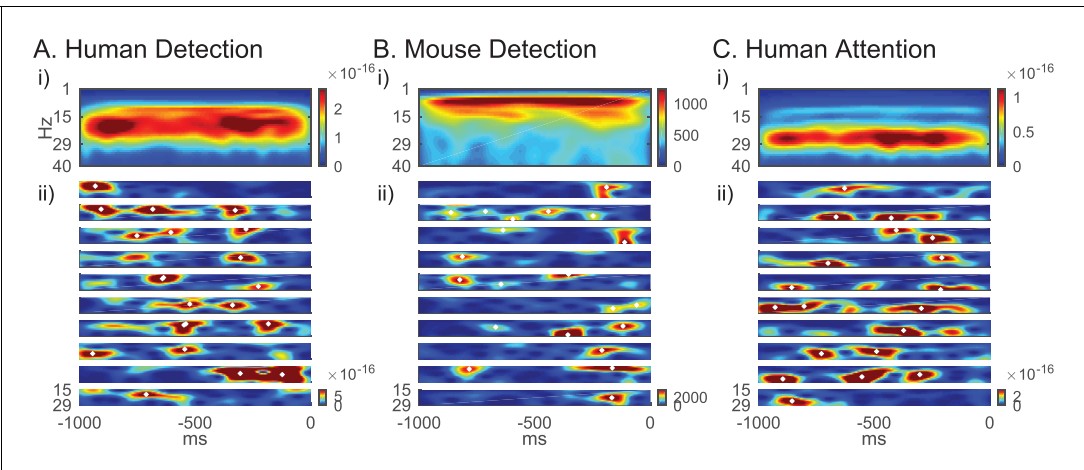

**Figure 2.** Beta emerges as brief events in non-averaged spectrograms. (**A**) Top panel shows averaged spectrogram (1–40 Hz) in the 1 s prestimulus period from 100 trials, from an example subject in the human detection dataset (average across 100 non-detected trials). The bottom panels show examples of prestimulus beta band (15–29 Hz) activity in 10 trials from the same subject. White asterisks denote local maxima in the spectrogram with maxima power above 6X median power of the maxima frequency. (**B**) Same format as A in mouse detection task (average across 130 non-detected trials), and (**C**) in human attention task (average across 100 attend out trials).
DOI: https://doi.org/10.7554/eLife.29086.004

Given that beta is event-like in non-averaged spectrograms, there are several possible features of such events that could contribute to differences in prestimulus beta power averaged across time and frequency. The features that could underlie a low versus high value of averaged prestimulus power (*Figure 3* top) include a difference in: The event number (i.e. rate); event power; event duration; and/or event frequency span (*Figure 3A–D*, respectively).

To quantify these high-power transient beta events and their features in the non-averaged spectral activity, we defined beta events as local maxima in the 1 s prestimulus spectrogram for which the maxima frequency fell within the beta band (15–29 Hz) and the maxima power exceeded a set power cutoff. To choose the power cutoff in a principled manner, we calculated for each subject and session the correlation between the percent area above cutoff in the 1 s prestimulus beta band spectrogram (see red in *Figure 4Ai* inset) and the trial mean prestimulus beta power (*Figure 4i*). The correlation with mean power was highest near the 6X median power cutoff across species and behavioral tasks (dotted line in *Figure 4i*). In other words, this power cutoff best accounted for the variability in trial mean prestimulus power, the effect we are seeking to understand. We therefore chose the value of 6X median as the cutoff for further analyses (see also figure supplements to Figures 6, 7 and 8 for examination of cutoff variation). The inverse cumulative density function (1-CDF) of all local maxima in the beta band as a function of the power cutoff shows that 6X median cutoff captures ∼20% of all local maxima (*Figure 4ii*).

The probability histogram of the number of events in the prestimulus period, and the event power, duration and frequency span, are plotted in *Figure 5*. Event power was defined as the spectrogram value at the maxima, and duration and frequency span were defined as full-width-at-half-maximum of the beta event in time and frequency dimensions, respectively. There was homology in the distribution of beta event features across tasks and species: In almost all trials,<4 beta events occurred in the 1 s prestimulus period; the event power probability histogram fell off monotonically after the power cutoff; the event duration was confined to a restricted range around a stereotypical value (mean ±SEM across events: human detection 167 ± 1.6 ms; mouse detection 145 ± 1.3 ms; human attention 153 ± 1.4 ms); and, the event frequency span was confined around a stereotypical value (mean ±SEM across events: human detection 8.6 ± 0.07 Hz; mouse detection 10.6 ± 0.11 Hz; human attention 8.9 ± 0.07 Hz). Below the 6X median threshold, event duration and frequency span become more distributed (*Figure 5—figure supplement 1*).

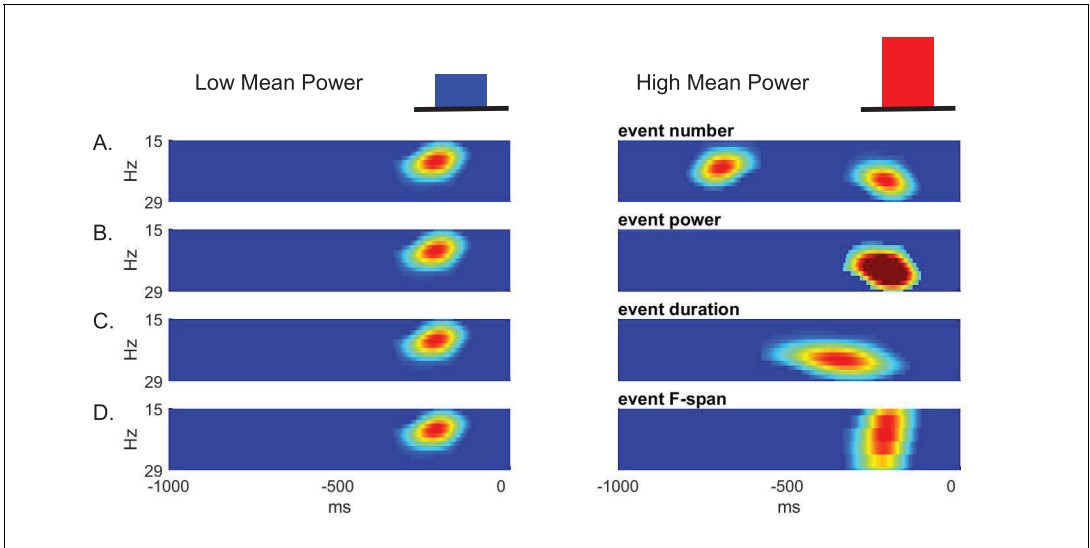

**Figure 3.** Schematic illustration of possible features underlying differences in averaged prestimulus beta power. Given that surges in beta power in non-averaged data occur as transient events, higher trial mean beta power could be due to an increase in (**A**) event number (i.e. rate), (**B**) event power, (**C**) event duration, and/or (**D**) event frequency span (F-span).
DOI: https://doi.org/10.7554/eLife.29086.005

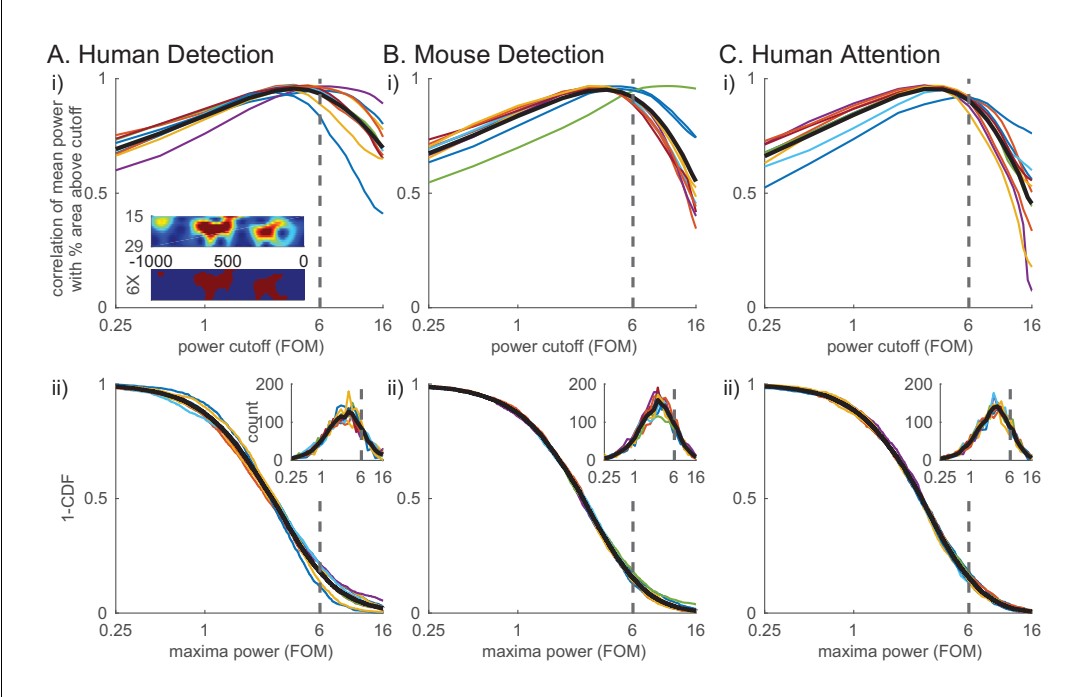

**Figure 4.** Beta events defined by a 6X median power cutoff show consistently high correlation with average prestimulus beta. (i) Pearson's correlation coefficient between mean prestimulus beta power and the percent area (i.e. percentage of pixels in the spectrogram) above cutoff in the non-averaged spectrogram, for various cutoffs calculated as factors of median and plotted on a log scale, in each dataset (A–C) (thin colored curves for individual subjects/sessions; thick black curve for average across subjects.). Inset in A shows an example spectrogram (top) with illustration of percent area above cutoff for a 6X median cutoff (below). (ii) Distribution of maxima power for all beta-band local maxima in non-averaged spectrograms. 1-CDF shows proportion of local maxima above the cutoff (thin colored curves for individual subjects/sessions; thick black curve for aggregate of all local maxima across subjects/sessions). Insets show the same data as histograms.

DOI: https://doi.org/10.7554/eLife.29086.006

## The rate of prestimulus beta events had the highest correlation with trial mean prestimulus power and was the most consistent predictor of behavior

Next, we assessed the impact of each beta event feature (number, power, duration, frequency span) on the trial mean prestimulus beta power, and on behavioral measures of perception and attention. *Figure 6* shows for all subjects and sessions in each dataset the Pearson's correlation between trial mean prestimulus power and the trial summary of each beta event feature in the prestimulus period (event number per trial, trial mean event power, duration and frequency span). Again, there was consistency across species and tasks. The correlation coefficient distributions showed significant differences in all three datasets (Friedman test: human detection $p = 3.92 \times 10^{-6}$, $df = 3$, $\chi^2 = 27.8$; mouse detection $p = 5.89 \times 10^{-6}$, $df = 3$, $\chi^2 = 27$; human attention $p = 2.33 \times 10^{-6}$, $df = 3$, $\chi^2 = 28.9$). In each dataset, the number of beta events in the prestimulus period (i.e. rate) showed a significantly higher correlation with trial mean prestimulus beta power than the trial mean event power, duration or frequency span (*Figure 6* asterisks denote p<0.05, Holm-Bonferroni corrected post-hoc Wilcoxon signed rank test; see also *Figure 6—figure supplement 1* for scatterplots of example subject data and *Figure 6—figure supplement 2* for variations in power cutoff).

The number of beta events in the prestimulus period was also the most consistent predictor of behavior across datasets. *Figure 7i* shows that prestimulus beta event rate mirrors the beta power (*Figure 1i*), where the event rate is higher in non-detected compared to detected trials throughout most of the 1 s prestimulus window. In *Figure 7ii*, we sorted the trials in ascending order per trial summary of each beta event feature in the prestimulus period (event number per trial, trial mean event power, duration and frequency span), and calculated the percentage of detected or attend-in

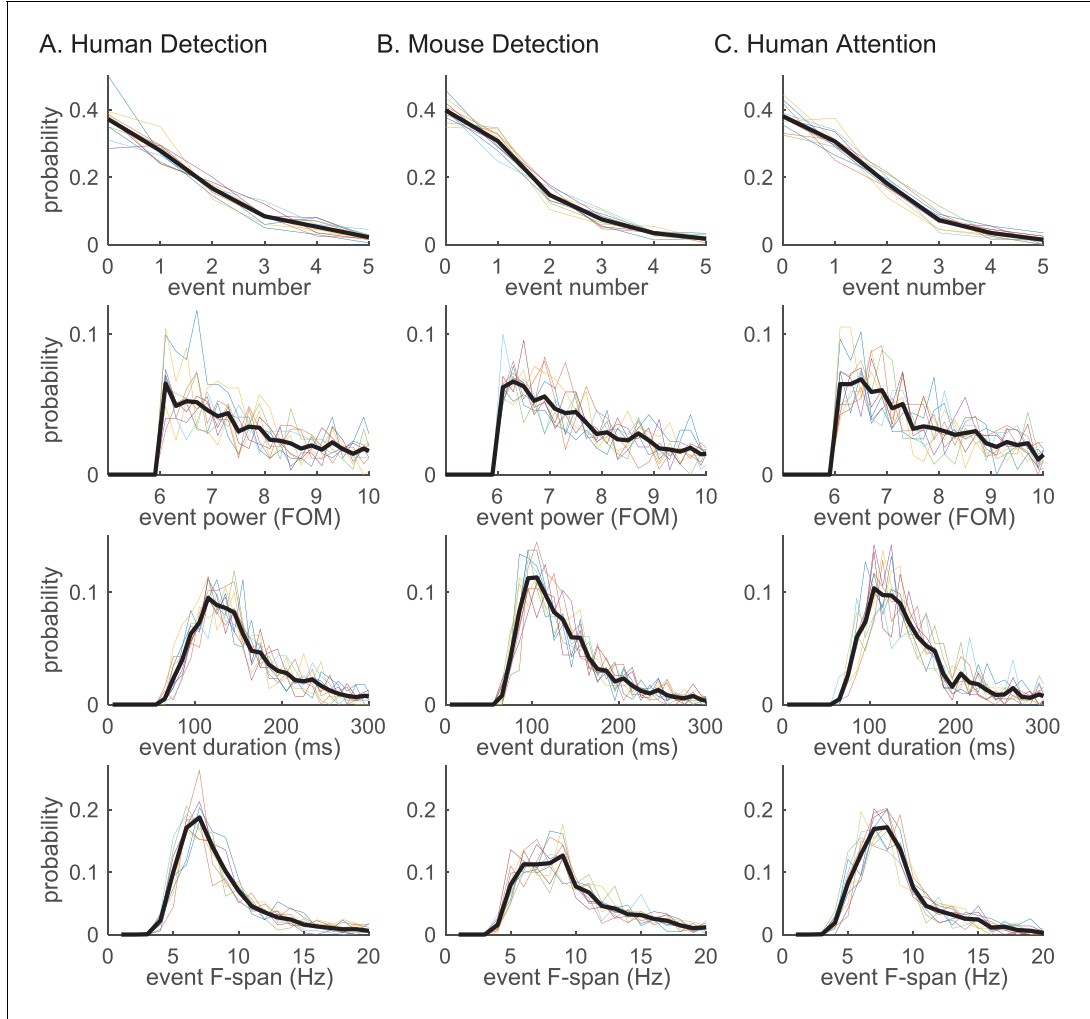

**Figure 5.** Beta event features are highly conserved across tasks and species. Probability histogram for each prestimulus beta event feature; (i) event number, (ii) event power, (iii) event duration, (iv) event frequency span; defined at 6X median cutoff in each dataset (**A–C**). Probability was calculated per trial for event number, and per event for event power, duration and frequency span. Probability histograms are plotted for each subject/session (colored); and for aggregate of all trials (for i) or events (for ii-iv) across subjects/sessions (black). Bin intervals for each histogram were as follows; 1 for event number; 0.2 (FOM) for event power; 10 (ms) for duration; and 1 (Hz) for frequency span.

DOI: https://doi.org/10.7554/eLife.29086.007

The following figure supplement is available for figure 5:

**Figure supplement 1.** Beta event duration and frequency span is stereotyped when maxima power is higher than 6X median cutoff.

DOI: https://doi.org/10.7554/eLife.29086.008

trials as PCM in 21-trial sliding bins (as in *Figure 1ii*). This visualization shows that the negative trend is more prominent in event number in all three datasets. To determine whether these negative relationships produced a significant difference across behavioral conditions, we compared the pooled average of each feature across conditions (*Figure 7iii*). In the human data, only the number of events per trial was significantly higher in non-detected (miss)/attend out conditions (*Figure 7iii*, asterisks $p<0.05$, right-tailed paired t-test), and only the number of events had DP/AP distributions that were significantly less than 0.5 across subjects (*Figure 7iv*, asterisks $p<0.05$, one-sample left-tailed Wilcoxon signed rank test). The number of beta events was also predictive of behavior in the mouse LFP. In this dataset, trial mean event power and frequency span were also predictive of non-detection, possibly due to the different signal to noise ratio stemming from the different recording modalities. However, in all three datasets, the effect size of the difference across behavioral conditions was higher for event number than any other event feature (see Cohen's d values in *Figure 7iii*). Further,

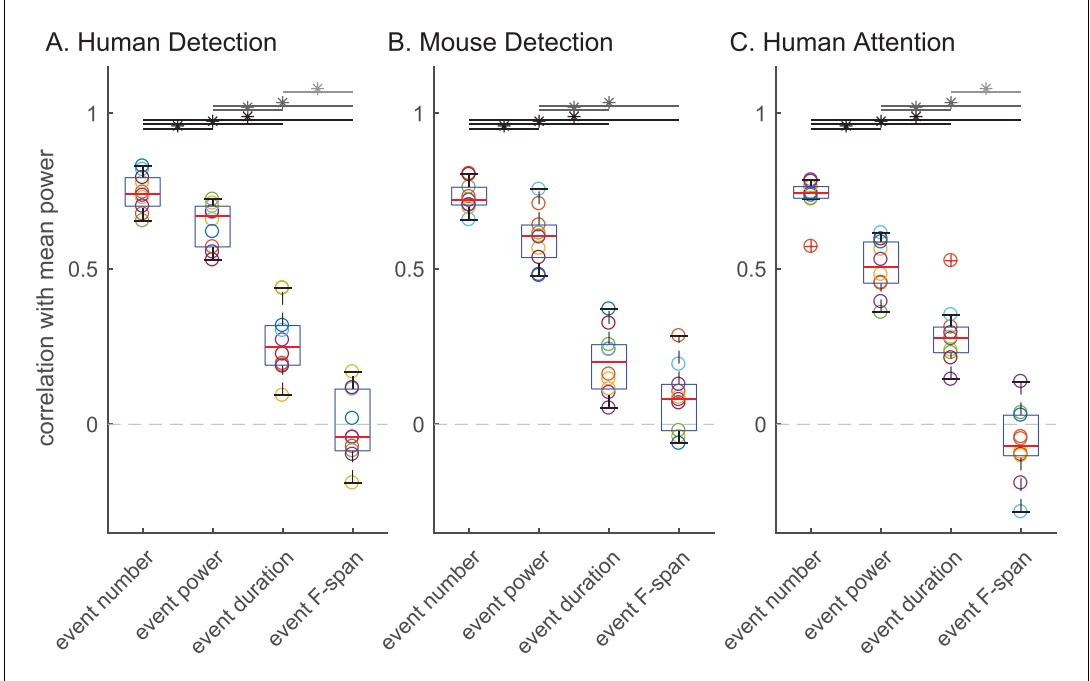

**Figure 6.** The number of beta events has a higher correlation with trial mean prestimulus beta power than event power, duration or frequency span. Box and whisker plots over subjects/sessions depicting the Pearson's correlation coefficients ($R$) between the trial summary of each beta event feature and trial mean prestimulus beta power. Note, for all analyses involving trial mean event power, duration and frequency span, only the trials with one or more events were considered. Friedman test followed by right-tailed Wilcoxon signed-rank test for each pair of event features, with Holm-Bonferroni correction for multiple comparisons; significance was determined at $p < 0.05$ (asterisks).
DOI: https://doi.org/10.7554/eLife.29086.009

The following figure supplements are available for figure 6:

**Figure supplement 1.** Example scatter plots showing relationship between trial mean prestimulus beta power and corresponding beta event number, trial mean event power, duration and frequency span.
DOI: https://doi.org/10.7554/eLife.29086.010

**Figure supplement 2.** Dependence of correlation with mean power on choice of cutoff.
DOI: https://doi.org/10.7554/eLife.29086.011

the percent of subjects/sessions that had DP/AP significantly less than 0.5 was highest for event number in all three datasets (as determined by 95% confidence interval, constructed from bootstrapping 1000 times; see parentheses *Figure 7iv*). Taken together, these results suggest that the event number (i.e. rate) was the most consistent predictor of behavior across tasks and species.

As stated above, we chose a 6X median power cutoff to define beta events because this value consistently showed the highest correlation between percent area (in spectrogram) above power cutoff and averaged prestimulus power (*Figure 4*). We investigated the impact of varying the power cutoff on the correlation between each beta event feature and trial mean prestimulus power, and on the relationship between each beta event feature and behavior. We found that as the beta event power cutoff was lowered, the trial mean event power became more strongly correlated with mean prestimulus power, while event number became less correlated (the two curves cross at ~3X median cutoff, *Figure 6—figure supplement 2*). This result might be expected from including a greater number of local maxima in the spectrogram that are within the beta frequency, but smaller in amplitude and closer to the signal to noise boundary. Similarly, event power was a stronger predictor of behavior than event number for low power cutoffs (~<3X median, *Figure 7—figure supplement 1*).

Given that event number was the most consistent predictor of detection/attention, we determined the number of events that best separated the behavioral conditions, i.e. the optimal event number criterion. *Figure 7—figure supplement 2i* displays the probability histogram of event number for each behavioral condition. These distributions and the corresponding effect size values in

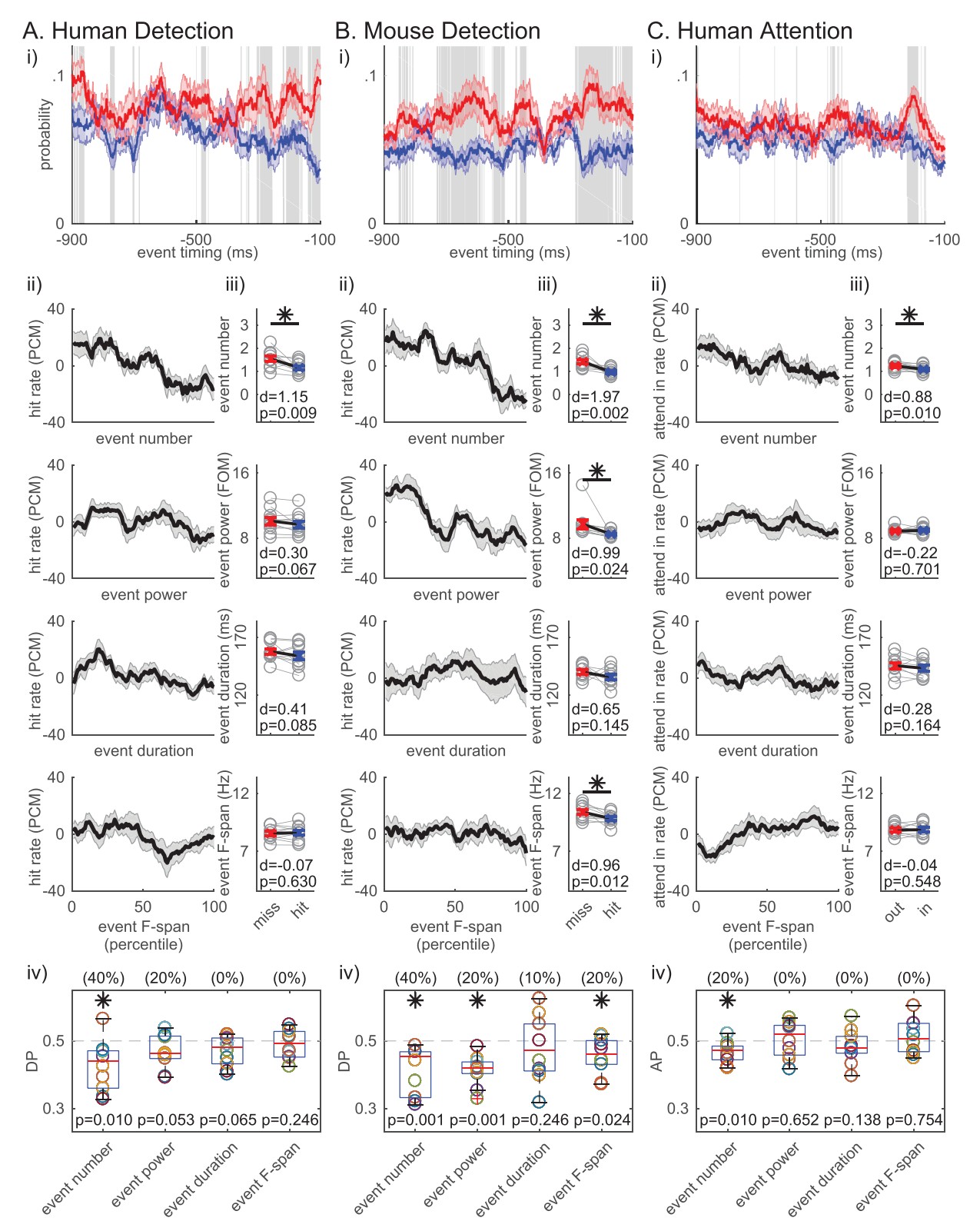

**Figure 7.** Beta event number per trial (i.e. rate) consistently predicts detection/attention across tasks and species. (i) Probability time histogram of beta event occurrence in the 1 s prestimulus window. Data binned in 50 ms windows, sliding in 1 ms steps. All other notations are same as *Figure 1i*. (ii-iv) Analogous analyses to *Figure 1ii-iv* were done for ii-iv, for each trial summary event feature (number, power, duration, frequency span) separately. Only trials with one or more events were considered for trial mean event power, duration and frequency span.

*Figure 7 continued on next page*

*Figure 7 continued*

DOI: https://doi.org/10.7554/eLife.29086.012

The following figure supplements are available for figure 7:

**Figure supplement 1.** Dependence of relationship between beta event features and behavior on choice of power cutoff.

DOI: https://doi.org/10.7554/eLife.29086.013

**Figure supplement 2.** Optimal criterion for classifying behavior based on event number is two events per trial.

DOI: https://doi.org/10.7554/eLife.29086.014

*Figure 7—figure supplement 2D* suggest that trials with zero events were more likely to be detected, and trials with two or more events were more likely not to be detected. Trials with zero events were slightly more likely in the attend-in condition, and trials with two or more events occurred more often in the attend-out condition.

Next, we considered a binary classifier that categorized trials as non-detected/attend out when event number was at or above a certain criterion value (trials categorized as detected/attend in when event number was less than the criterion). The optimal criterion for each subject/session was the criterion at which the binary classifier achieved best separability between its true positive rate and its false positive rate. Consistent with the distributions and effect sizes, in each dataset, the optimal event number criterion was two events for most subjects/sessions (*Figure 7–figure supplement 2ii*; 60% of subjects in human and mouse detection, and 50% of subjects in human attention). Put another way, trials with $\geq 2$ prestimulus events predict non-detection/attend out conditions and trials with 0 events predict detection/attend in conditions, while trials with one event are not predictive without additional information about event features.

## Non-detected trials were more likely to have a prestimulus beta event closer to the time of the stimulus

We next asked whether the event timing relative to the stimulus onset may influence detection, such that the closer in time the beta event is to the sensory stimulus, the more likely for the stimulus to not be detected. We found that in trials with one event, the event timing was significantly closer to stimulus onset on non-detected trials across human detection subjects, and trended towards significance (p<0.1) in the mouse detection sessions (right-tailed paired t-test, *Figure 7–figure supplement 2iii, A and B*). We would not expect a difference in timing relative to the stimulus onset in the different attention conditions, as the timing of the attentional cue relative to the stimulus onset was randomized (confirmed in *Figure 7–figure supplement 2Ciii*).

Based on these results, we hypothesized that, for any number of events in the prestimulus period, the timing of the beta event occurring closest to stimulus onset (i.e. 'most recent beta event') mattered for detection. The probability time histogram of the 'most recent beta event' per trial showed that beta events were more likely to happen closer to stimulus onset on non-detected trials (*Figure 8i*). The probability time histogram profiles were significantly different out to ~200 ms prestimulus in the human and mouse data, suggesting that beta events' influence on function may have an effective period that lasts about ~200 ms (pointwise left-tailed Wilcoxon signed rank test, p<0.05).

We further assessed the relationship between detection performance and features of the 'most recent beta event', including its timing relative to stimulus onset, and its power, duration and frequency span. Detection performance was quantified with the same measures as in *Figure 7*. In the human data, the 'most recent beta event' timing (relative to the stimulus onset) was the only significant feature, such that the closer the 'most recent beta event' was to the stimulus the less likely the subject was to detect (*Figure 8Aii-iv*; right-tailed paired t-test comparing the pooled average between behavioral conditions, and one-sample left-tailed Wilcoxon signed rank test of the DP distribution). In the mouse data, the timing of the 'most recent beta event' predicted detection as well (*Figure 8Bii-iv*). As in *Figure 7Biii-iv*, 'most recent beta event' power and frequency span also showed significance in the mouse data. However, the effect size was consistently highest for 'most recent beta event' timing in humans and mice (*Figure 8iii*), and the percent of subjects/sessions with DP significantly less than 0.5 was highest for 'most recent beta event' timing (as determined by 95% confidence interval, constructed from bootstrapping 1000 times; see parentheses *Figure 8iv*).

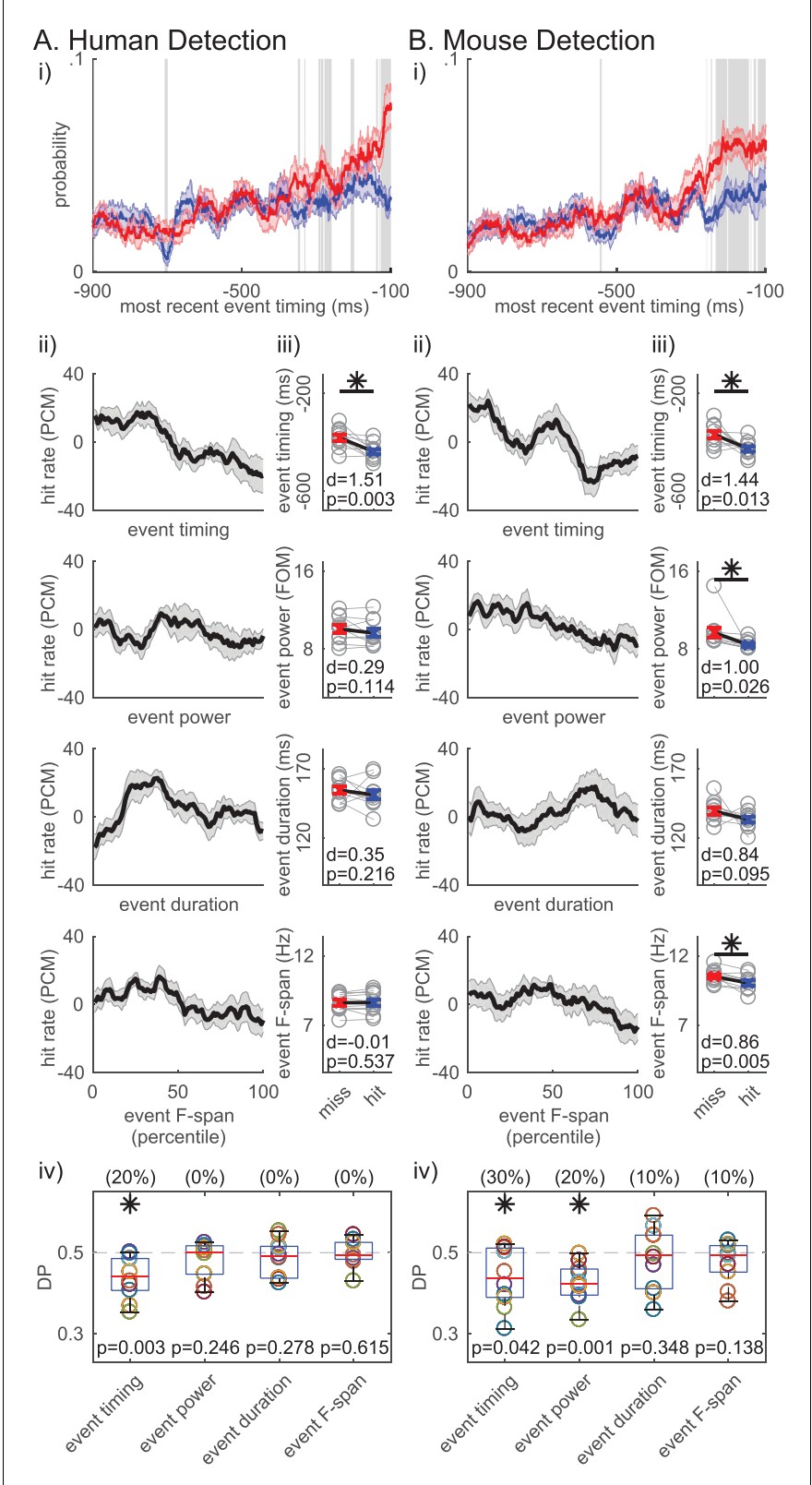

**Figure 8.** Non-detected trials are more likely to have a beta event closer to the time of the stimulus. For each trial, the event occuring closest to the stimulus onset was termed the 'most recent beta event'. All analyses are analogous to *Figure 7*, in the human (A) and mouse (B) detection datasets, but for features of the 'most recent

*Figure 8 continued*

beta event' (event timing relative to the stimulus, event power, duration and frequency span). For analyses in ii-iv, only the trials with one or more events were considered.

DOI: https://doi.org/10.7554/eLife.29086.015

The following figure supplement is available for figure 8:

**Figure supplement 1.** Dependence of relationship between 'most recent beta event' features and behavior on choice of cutoff.

DOI: https://doi.org/10.7554/eLife.29086.016

Hence, the timing of the 'most recent beta event' influenced detection more consistently than its power, duration or frequency span.

Variation of the power cutoff defining beta events for features of the 'most recent beta event' revealed similarities between timing and event number. As the cutoff was lowered below ~3X median, the 'most recent event' power had a more significant influence on perception than the timing relative to the stimulus onset (*Figure 8—figure supplement 1*).

Given that both the event number and the 'most recent event' timing significantly impact function, we next assessed whether these variables are dissociable in their influence on detection. To address this question, we first assessed whether 'most recent event' timing affected detection independent of event number by analyzing whether the 'most recent event' timing was predictive of detection in event number matched trials. This matching across behavioral conditions was achieved via a random trial trimming process, where the histograms of event number were forced to be identical in detected and non-detected trials (*Figure 9i* left panels, black curves). The event number matching was restricted to trials with at least one event, to be able to compare 'most recent beta

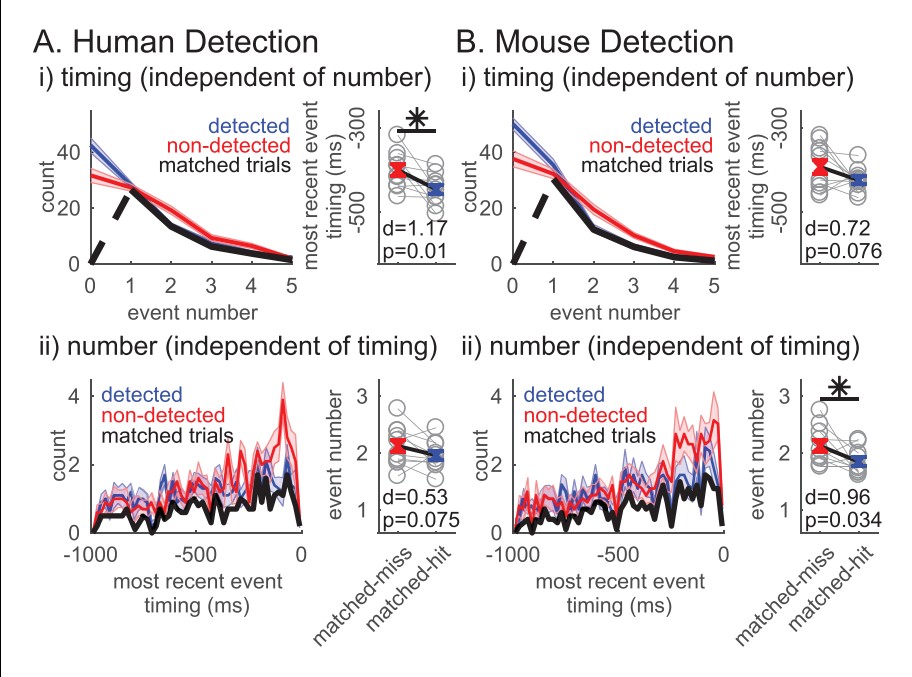

**Figure 9.** Independent influence of rate and timing on detection. (i)Left panel: Event number histogram in detected (blue, mean ± SEM), non-detected (red, mean ± SEM), and event number matched (black, mean across subjects/sessions) trials via a random trial trimming process. Right panel: Pooled average of 'most recent event' timing on event number matched non-detected and detected trials. Cohen's d and p-value for right-tailed paired t-test denoted, asterisk p<0.05. (ii) Analogous to i, where 'most recent event' timing histograms (20 ms bins) were matched via a random trial trimming process (left panel, black). Pooled average of event number on 'most recent event' timing matched trials.

DOI: https://doi.org/10.7554/eLife.29086.017

event' timing. Approximately 50% of trials remained after matching (mean ±SEM across subjects/sessions: human detection 52.4 ± 1.5%; mouse detection 48.1 ± 1.6%). The pooled average of 'most recent beta event' timing was significantly closer to stimulus onset on non-detected trials in the human detection dataset (right-tailed paired t-test, p=0.01) and trended toward significance in the mouse detection dataset (p=0.076). These results provide support for the hypothesis that the 'most recent beta event' timing can be predictive of detection independent of event number.

We similarly assessed whether event number affected detection independent of the timing of the event closest to stimulus onset, that is the 'most recent event' timing. The 'most recent event' timing histogram in trials with at least one event was forced to be the identical via an analogous random trial trimming process (*Figure 9ii* left panels black curves, matched in bin resolution of 20 ms). After matching, approximately 33% of all trials remained (mean ±SEM across subjects/sessions: human detection 34.5 ± 2.3%; mouse detection 32.4 ± 0.9%). In the mouse dataset, the pooled average of event number was significantly higher in the non-detected trials (right-tailed paired t-test, p=0.034), and trended towards significance in the human detection dataset (p=0.075). Hence, these results provide support for the hypothesis that event number can be predictive of detection independent of event timing.

## Beta events did not occur rhythmically

We next assessed whether beta events occurred rhythmically in the prestimulus period, and if the degree of rhythmicity impacted behavior. For this purpose, we quantified the inter-event interval (IEI) as the difference in timing between two consecutive events, in trials that had two or more events (~30% of trials). The IEI distributions did not differ between detected versus non-detected or attend-in versus attend-out conditions (*Figure 10i*). Moreover, there was no clear evidence in the IEI distribution that beta events occur rhythmically in 1 s prestimulus windows. The lack of rhythmicity was further supported by analyses of the Fano Factor and squared coefficient of variation ($CV^2$). The Fano Factor quantifies the trial-to-trial variability in event number (i.e. rate), and the $CV^2$ quantifies

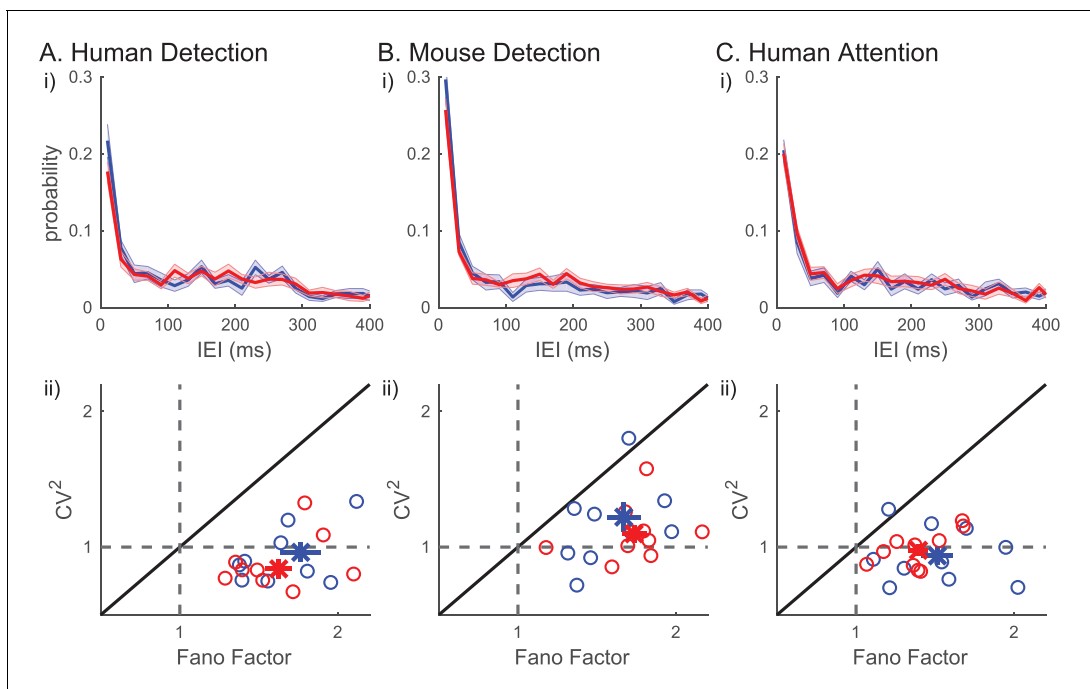

**Figure 10.** Inter-event-interval (IEI) analyses suggest that beta events are not rhythmic or a renewal process. (i) IEI distributions (20 ms bins) on detected/attend in (blue) and non-detected/attend out (red) conditions; mean across subjects with ± SEM as error bars. (ii) Fano factor and $CV^2$ for each subject are plotted as O, for all trials (black), for only the detected/attend in trials (blue); and only the non-detected/attend out trials (red). The mean across subjects are plotted as x, with ± SEM as error bars.

DOI: https://doi.org/10.7554/eLife.29086.018

the variability in IEI. For a Poisson process, the Fano Factor and $CV^2$ would be 1, whereas for a more regular (rhythmic) process, both values would be less than 1. In all three datasets, $CV^2$ was around 1 (*Figure 10ii*). The $CV^2$ distributions were not significantly different from one across human subjects (two-tailed Wilcoxon signed rank test: human detection p=0.106; human attention p=0.375) and trended towards being higher than one across mouse detection sessions (p=0.065). The Fano Factor values were above one in all subjects/sessions across all three datasets (*Figure 10ii*, one-sample right-tailed Wilcoxon signed rank test: $p = 9.77 \times 10^{-4}$ for all three datasets). To further investigate the characteristics underlying the beta event generation process, we looked at the relationship between the Fano Factor and $CV^2$. A renewal process, defined as a process by which the time of occurrence of an event only depends on the time of occurrence of the previous event, would have similar values for Fano Factor and $CV^2$. However, in each dataset, almost all data points had Fano Factors larger than $CV^2$, indicating that the driver underlying beta event generation was not a renewal process.

*Figure 10i* shows that many of the IEIs were concentrated at <150 ms, while a typical event duration was ~150 ms (*Figure 5*). This result suggests there were cases where multiple events occurred within one continuous high-power region (i.e. above the power cutoff, see *Figure 2ii* for examples). To determine whether multiple events happening in quick succession (i.e. multi-maxima events within one continuous suprathreshold region) constituted a separate class of events with a different functional meaning, we considered an alternative definition of events. Here, each continuous supra-cutoff temporal region was defined as an event, and they were referred to as 'non-overlapping beta events' (*Figure 11*). Most supra-cutoff regions had a single local maximum (~80%) and the probability of a given number of maxima per event was not significantly different across behavioral conditions (*Figure 11i*). This suggests that multi-maxima events did not constitute a distinct class of events, with respect to how they influenced behavior.

Successive supra-cutoff temporal regions were by definition separated by periods below the power cutoff, which effectively created a refractory period in the IEI histogram of 'non-overlapping events' (*Figure 11ii*). Likely due to this refractory period, the IEI histogram started at zero and rose to a peak near 200 ms. An alternative possibility is that the peak near 200 ms is reflective of underlying rhythmicity. To test the rhythmicity of 'non-overlapping events', we constructed jittered distributions of IEIs (*Figure 11ii*, black curve; see legend and Materials and methods for details). Neither the detected nor the non-detected (and, attend-in nor attend-out) IEI distributions were distinct from the jittered distributions, suggesting that 'non-overlapping beta events' did not occur rhythmically in any behavioral condition. This further corroborates the conclusion that beta event generation is not rhythmic. Note, with both beta event definitions, conclusions on rhythmicity are limited by our restriction to the 1 s prestimulus period.

Results on the relationships between beta event features and power and behavior also held under the 'non-overlapping events' definition, such that event number showed the highest correlation with mean prestimulus power (*Figure 11iii*) and was the most consistent predictor of detection/attention (*Figure 11iv*). This high congruency can be attributed to the fact that, as stated above, in all three datasets ~ 80% of non-overlapping events have only one local maximum, meaning they were also considered to be an event in the original definition.

## Beta events are generated by a bursty mechanism, rather than by dynamic amplitude modulation of a sustained beta oscillation

The lack of evidence for rhythmicity of beta events (*Figure 10i* and *Figure 11ii*), together with the Fano Factors being well over 1 (*Figure 10ii*), implies that the underlying mechanism generating beta events was less regular than a Poisson process and more akin to a 'bursty' process. We investigated this possibility further by considering two mechanisms that could underlie transient surges in beta power in the spectrogram (*Figure 2*), a Dynamic Amplitude Modulation mechanism versus a Bursty Generator mechanism (*Figure 12* schematic). A Dynamic Amplitude Modulation hypothesis predicts the existence of an ongoing continuous rhythm whose amplitude (and hence power) was dynamically modulated. Under this hypothesis, the rhythmicity of the beta component of the signal would be independent of its amplitude. In contrast, the Bursty Generator hypothesis assumes that the generator that triggers the beta waveform also causes the transient surge in beta amplitude. Under this hypothesis, beta rhythmicity is only present when this generator triggers the system: If so,

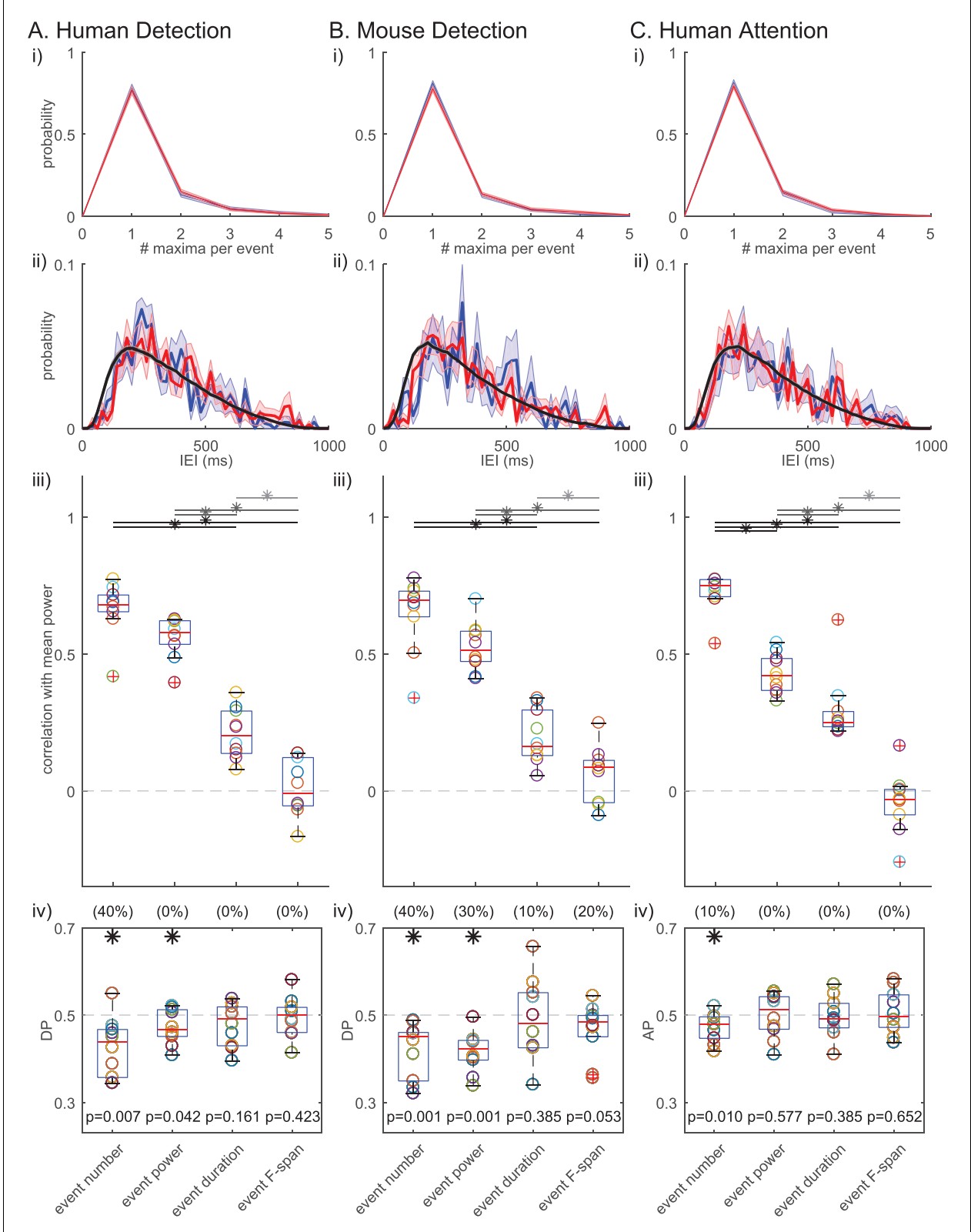

**Figure 11.** Beta event findings are robust under redefinition of events as 'non-overlapping beta events'. (i) Probability histograms for number of local maxima per supra-cutoff region (i.e. a 'non-overlapping beta event'), in detected (blue) and non-detected (red) trials; mean ± SEM across subjects. (ii) IEI distributions (20 ms bins) on detected/attend in (blue), and non-detected/attend out (red) conditions. The jittered distribution (black) was generated

*Figure 11 continued on next page*

*Figure 11 continued*

by randomly jittering the event timing within the 1 s prestimulus window 1000 times. Mean across subjects/sessions. (iii) As in *Figure 6*; and, (iv) As in *Figure 7iv* for 'non-overlapping beta events.'.

DOI: https://doi.org/10.7554/eLife.29086.019

rhythmicity would be restricted to high beta power regions of the data. The contrasting predictions of these two competing hypotheses were tested by choosing an amplitude cutoff, and comparing the rhythmicity in time lags above and below the cutoff.

The intuition behind this analysis is shown in *Figure 12A*. As diagrammed, we assessed whether or not rhythmicity was maintained by comparing the time lag between first and last peaks in the oscillation within a high power region, or last and first peaks in consecutive regions with a low power region in between. For this analysis, we band-passed the data and performed a Hilbert transform to calculate the time lag modulo the beta period for these regions. These values were plotted as functions of the normalized modulo (remainder after division) that varied between −0.5 and 0.5, where 0 corresponded to integer multiples of the beta period. The 'time lag above cutoff' (red periods in *Figure 12A*) were calculated as the time between the first (orange dot) and last (brown dot) 0° phase

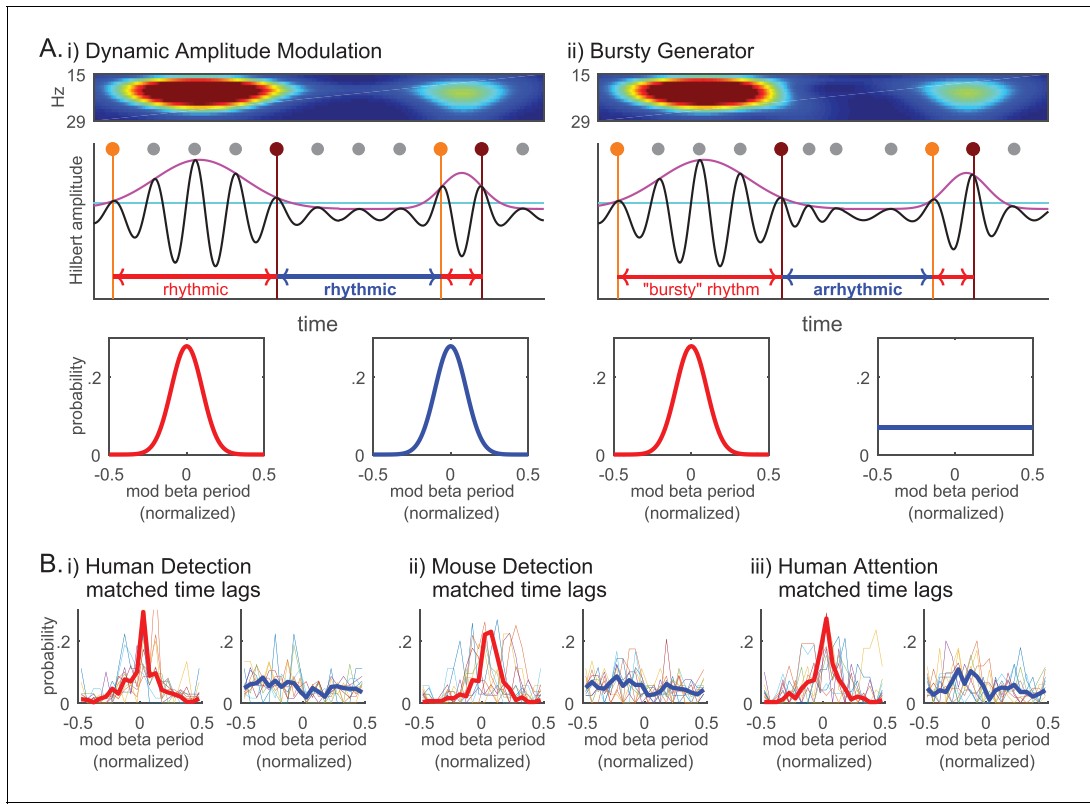

**Figure 12.** Beta event emergence is consistent with a bursty generator mechanisms, as opposed to dynamic amplitude modulation of a sustained beta oscillation. (**A**) Schematic illustration of two alternative mechanisms that could underlie transient surges in beta power in the spectrogram: (i) Dynamic Amplitude Modulation mechanism; versus, (ii) Bursty Generator mechanism. Hilbert amplitude envelope in magenta and the amplitude cutoff in cyan. For each supra-cutoff region, orange and brown dots denote the first and last peaks in the oscillation (phase 0° time points), respectively; all other peaks are denoted with grey dots. The Dynamic Amplitude Modulation hypothesis would predict that the probability histogram of time lag modulo the beta period is centered around zero for both the time lags above (lower left panel, red curve) and below (lower right panel, blue curve) the cutoff, indicating rhythmicity in each region. In contrast, the Bursty Generator hypothesis would predict that the modulo probability histogram is centered around zero for the time lags above the cutoff (red), but flat for the time lags below the cutoff (blue) that lacks rhythmicity. (**B**) The probability histograms of time lag modulo the beta period are compared for the regions above (left panels, red) and below (right panels, blue) the 2X median amplitude cutoff (thin colored curves for individual subjects/sessions; thick red and blue curves for aggregate of all subjects/sessions in each dataset).

DOI: https://doi.org/10.7554/eLife.29086.020

time points in each supra-cutoff region, where 0° phase denotes the peak points in the oscillation (marked with grey/orange/ brown dots in *Figure 12A* schematic). The 'time lag below cutoff' was calculated as the time between the last 0° phase time point in the preceding supra-cutoff region and the first 0° phase time point in the succeeding supra-cutoff region. With this definition, we would expect the time lags to be integer multiples of the beta period in rhythmic regions, i.e. we would expect the modulo of the beta period to be around 0. Under the Dynamic Amplitude Modulation hypothesis, where a sustained oscillation is being modulated in amplitude, we would expect both the time lags above (red) and below (blue) the cutoff to be rhythmic, hence the time lag modulo the beta period distributions should both be centered at 0 (lower panels in *Figure 12Ai*). In contrast, the Bursty Generator hypothesis would predict that supra-cutoff regions would be rhythmic, whereas sub-cutoff regions would not. In this model, the time lag modulo the beta period distributions would be centered at 0 for time lags above cutoff (red), whereas the distribution would be flat for time lags below cutoff (blue) (lower panels in *Figure 12Aii*).

We chose the Hilbert-amplitude cutoff (cyan horizontal line in *Figure 12A*) such that it was comparable to the 6X median power cutoff in the Morlet spectrogram analysis, in that there was a similar percentage of time points above the cutoff in the beta band (~10%). We determined the characteristic beta period for each subject/session as the inverse of the characteristic beta frequency. The characteristic beta frequency in turn was defined as the median of the instantaneous frequency values in the Hilbert-transformation of the beta band-passed data. This characteristic beta period was used to calculate the time lags' modulo of the beta period and to normalize to range between −0.5 and 0.5.

To account for the possibility that rhythmicity may decay over time, we first performed histogram matching of time lags above and below cutoff, such that the number of beta cycles in the time lags being compared was matched. The modulo probability histograms for the matched time lags above and below the amplitude cutoff are plotted in *Figure 12B* for each dataset (i-iii). These probability histograms show that time lags above cutoff are integer multiples of the beta period (i.e. distributions are centered at 0), in direct contrast to time lags below cutoff that are not (i.e. distributions are flat).

Overall, these results strongly support the prediction that high power regions are rhythmic but low power regions are not, providing evidence for the Bursty Generator hypothesis over the Dynamic Amplitude Modulation hypothesis

## Discussion

Our data showed that averaged prestimulus beta activity in somatosensory neocortex is predictive of perceptual success and shifts in attention, such that higher power reflects decreased detection or attention. This result held across various analysis methods, species and recording modalities (human MEG, mouse LFP), consistent with the general prediction that spontaneous sensorimotor beta acts as, or is the by-product of, an 'inhibitory' process (*Engel and Fries, 2010*; *Haegens et al., 2011*; *Jones et al., 2010*; *Linkenkaer-Hansen et al., 2004*).

To further uncover the potential mechanisms underlying this robust association of beta with an 'inhibitory' influence, we studied the trial-by-trial relationship between beta and behavior. On individual trials, beta emerged as a brief high-power transient above background noise in the frequency spectrum and was consistent with a 'bursty' process as opposed to dynamic amplitude modulation of a sustained beta oscillation. As such, we postulated that observed differences in averaged power across behavioral conditions could reflect modulation in the number of high-power beta events in a predefined window (i.e. rate) and/or modulation in the event power, duration, or frequency span. Our results showed that the rate of prestimulus beta events was the most consistent predictor of detection and shifts in attention, and most strongly correlated with average prestimulus beta power. In human data, only the rate of events was significantly predictive of function across detection and attention tasks. Moreover, an event occurring closer to the stimulus onset was more likely to result in non-detection. Analysis of a subset of the data where the number of events, or conversely the timing of the 'most recent beta event', were matched across detected and non-detected trials showed that event number and timing may be independently informative to perception. Taken together, our findings indicate that with shifts in attention, the brain decreases the probability of high-power beta events, and this decrease translates to improved sensory processing in the aligned sensory channel. To our knowledge, these are the first results to show that rate of beta events in human neocortical

signals is a meaningful predictor of function. Prior studies in animal models have shown that features of transient beta activity in LFP signals correlate with behavior, including motor and cognitive (*Bartolo and Merchant, 2015*; *Feingold et al., 2015*) and working memory processes (*Lundqvist et al., 2016*). Homology in beta event characteristics across tasks, recording modalities, and species, suggests modulation of beta event rate may be the fundamental feature underlying behaviorally relevant differences in averaged beta power.

These observations have direct implications for the highly debated topic of whether there is a role for brain rhythms in information processing (*Ainsworth et al., 2012*; *Bastos et al., 2015*; *Engel et al., 2001*; *Lisman and Jensen, 2013*; *Merker, 2013*; *Ray and Maunsell, 2010*; *Ray and Maunsell, 2015*). A well-established theory suggests a crucial role for the specific alignment of oscillatory phases with external input, with specific phases beneficial for signal transmission while others are not (*Bastos et al., 2015*; *Lakatos et al., 2007*; *Schroeder et al., 2008*; *van Elswijk et al., 2010*). In contrast, our current study suggests that in spontaneous, prestimulus, anticipatory states it is unlikely that beta's local influence on the filtering of sensory input is through a temporal phase-alignment code. Rather, in all three of our tasks, the precise timing of the sensory stimulus onset is unpredictable for the subject (i.e. randomized design). Under these conditions, our results suggest that the inhibitory correlation between beta and perception may reflect the presence of another process that enhances beta event probability/rate, such as an increase in a specific thalamic input and/or the existence of a specific neuromodulatory environment that promotes this rhythm. Further, if a single beta event occurs within 200 ms before the sensory stimulus, the stimulus is less likely to be perceived, suggesting that the increased rate in non-attended conditions acts to increase the probability that an event will occur close to, or perhaps during, the stimulus. In this case, beta events could have an inhibitory impact on function for a prolonged period that extends longer than the event duration, a period that could be as long as 200 ms. Of note, this finding does not preclude a role for band-band phase-coherence between brain regions in information processing (*Sacchet et al., 2015*). Rather, our results are relevant to the potential interpretation of phase of a local spontaneous beta 'oscillation' at the timing of external stimulus arrival and suggest that the beta's inhibitory influence is likely not due to a specific phase-alignment.

How might an increase in the rate of prestimulus SI beta events translate mechanistically to a decrease in tactile perception? Our findings directly indicate that the circuit-level mechanisms that create beta events recruit an 'inhibitory' process that decreases the relay of sensory information to or from SI. The rate of beta events therefore predicts the presence of this inhibitory process at the time of the stimulus. Evidence for this inference depends first on understanding the mechanisms of generation of beta events observed in SI. We have previously investigated the mechanisms of high-power SI beta event generation (*Sherman et al., 2016*). Converging evidence in humans and animals supported a model-derived theory that the circuit-mechanisms creating high-power beta events consisted of bursts of subthreshold excitatory synaptic drive targeting proximal and distal dendrites of pyramidal neurons, such that the distal drive was effectively stronger and lasted ~50 ms. Our present study implies that beta's inhibitory impact on function is largely related to the likelihood of observing this strong distal drive (event number) and/or the strong distal drive happening closer to the onset of the stimulus ('most recent event' timing), in contrast to stronger distal drive (event power), longer lasting drive (event duration) or drive covering a larger frequency span in the spectrogram (event frequency span).

There are several plausible mechanisms by which the accumulation and/or the timing of the strong distal drive required for beta generation could impair throughput in the aligned somatotopic area. First, the strong distal drive to SI could decrease the efficacy of information processing by supragranular inputs to SI. One prominent candidate for this mechanism is recruitment of inhibitory neurons in supragranular layers (*Jones et al., 2009*). Such inputs could also alter distal dendritic processes in large layer V neurons (*Larkum, 2013*; *Murphy et al., 2016*; *Takahashi et al., 2016*), the primary contributor to the MEG signals recorded. Alternatively, the generators of the exogenous drive contributing to beta emergence in SI (e.g. bursting in thalamic neurons) could themselves mediate beta's inhibitory impact on perception at the locale of the exogenous generator (e.g. through a filtering process in the thalamus), in which case the increase in neocortical beta events represents an epiphenominal reflection of this process. Future studies are necessary to test alternative hypotheses and to identify possible cortical or thalamic sources of the exogenous drive.

Based on the following, we conjecture that the mechanisms of generation of high-power beta events may be distinct from lower-power beta transients, and thus separable in their impact on function. While the distribution of power of all beta local maxima did not show bimodality (*Figure 4ii* inset), several other factors indicated high power beta events were distinct from noise. First, high power beta events were stereotyped in their waveforms (*Sherman et al., 2016*) and in their spectral features, i.e. the event duration and frequency span (*Figure 5—figure supplement 1*). Second, sliding power cutoff analyses showed that despite the lack of clear bimodality in the full distribution of beta event power, there was a clear switch in the relative importance of event features as the beta event power cutoff was varied. For lower power cutoffs (<~3X median) beta event power is more predictive of averaged prestimulus power and behavior than beta event rate, while for higher cutoffs (>3X median) beta event rate is more predictive. Third, we found that the intervals between consecutive high power beta events were consistent with a bursty process, as opposed to dynamic amplitude modulation of a continuous process. Taken together, our prior (*Sherman et al., 2016*) and current results suggest that the mechanisms underlying the high-power beta events that dominate fluctuations in the averaged prestimulus beta power are distinct in their generation and relationship to function.

An open question is whether properties of transient rhythms in other frequency bands or subbands of beta may underlie observed differences across behavioral conditions. Indeed, transient rhythms emerging for a few cycles have been reported in other bands, including gamma (*Burns et al., 2011*; *Greenwood et al., 2015*; *Lundqvist et al., 2016*) and alpha (*Jones et al., 2009*; *Sherman et al., 2016*; *Ziegler et al., 2010*). In our human data, power-spectral density analysis shows a single 'bump' in the alpha band (7–14 Hz) and a single bump in the beta band (15–29 Hz); whereas the gamma (30–80 Hz) power is not distinguishable from the $1/f^n$ trend (*Jones et al., 2009*). The bump in the beta band emerges primarily between 18–22 Hz, also known as the beta 2 range, and we do not observe a separate bump for the beta 1 range (typically ~15 Hz; [*Kopell et al., 2014*]). We conjecture that this isolated increase in the power spectral density in the beta band is indicative of a singular underlying mechanism. In further support, as discussed above, we have previously shown evidence for a unique mechanism that generates stereotyped and homogeneous beta-band activity (*Sherman et al., 2016*). As a follow-up, the current study assessed the behavioral impact of beta's transient manifestation in detail. Of note, our data also show high power alpha band activity (*Jones et al., 2010*; *Jones et al., 2009*). We have observed that alpha and beta have separable correlations with attention and perception. Shifts in averaged alpha power occur sooner after attentional cues than shifts in beta power, and beta showed a stronger linear relationship to detectability (*Jones et al., 2010*; *Sacchet et al., 2015*). A full quantification of the potential transient nature of alpha and its relationship to beta events would be an interesting future direction.

The transient nature of high-power spontaneous beta events observed here has direct implications for brain stimulation studies, such as transcranial alternating current (tACS) or magnetic stimulation (TMS), aimed at entraining 'rhythms' to causally modulate behavior. Causal manipulations may be more effective if they are designed to match the intermittent character and specific temporal characteristics of the beta events observed. Our results suggest the rate and timing of beta events is a key target for modulation. Specifically, stimulation protocols that drive two or more events in the 1 s prestimulus period may impair perception, whereas blocking such events may benefit perception. Our data predict that driving single events could impair throughput in the aligned somatotopic area, but only if it arrives close to the time of the stimulus.

In summary, cumulative evidence across tasks, recording modalities, and species support the notion that as opposed to phase-alignment, amplitude modulation, sustained rhythmicity or several other possible means by which transient, high-power beta events could influence behavior (e.g. frequency span), beta event rate has a dominant impact on function.

## Materials and methods

### Human recordings
#### MEG data collection
Details of the human MEG recordings and source localization for both the detection and attention tasks have been previously reported (*Jones et al., 2010*; *Jones et al., 2007*). In brief, we used the

306-channel Vectorview system. For both the detection and attention dataset signals were sampled at 600 Hz with the band-pass set to 0.01 to 200 Hz. Dipole activity from a suprathreshold level stimulus to the third digit of the right hand was localized to the postcentral gyrus in the contralateral hand area of SI using least-squares fit inverse methods and individualized structural MRIs, or a spherically symmetric conductor model of the head (*Hämäläinen and Sarvas, 1989*). All analyses presented in this paper are derived from the forward solution from this source-localized region of interest. The evoked response elicited by the brief tap to the finger was reported in a prior report (*Jones et al., 2007*) and reproduced in *Figure 1—figure supplement 1Ai*.

### Detection task

Subject recruitment, experimental protocol, and data acquisition have been described in prior reports from our group (*Jones et al., 2010*; *Jones et al., 2007*). In brief, the stimulus was a single cycle of a 100 Hz sine wave (i.e. 10 ms duration) generated by piezoelectric benders (Noliac). The stimulus was applied to the third digit fingertip of the right hand. Individual subjects' perceptual thresholds were obtained before imaging by employing a parameter estimation by sequential testing (PEST) convergence procedure (*Dai, 1995*; *Leek, 2001*), which estimated the threshold to <5 µm precision.

During MEG imaging, 70% of the trials were maintained at perceptual threshold (50% detection) using a dynamic algorithm. 10% suprathreshold stimuli (100% detection) and 20% null stimuli (catch trials) were randomly interleaved with the threshold stimuli. Trial duration was 3000 ms. Trial onset was indicated by a 60 dB, 2 kHz auditory cue delivered to both ears for a duration of 2000 ms. The stimulus was delivered at a random time between 500 and 1500 ms after the onset of the auditory cue. The subjects were given 1000 ms to report detection or non-detection of the stimulus with button presses using the second and third digits of the left hand, respectively.

10 subjects underwent eight runs, 120 trials each, in one data collection session. To minimize within-session training effects, we limited our analysis to the last 100 trials of perceived and nonperceived threshold level stimuli for each subject (*Wan et al., 2011*).

### Cued attention task

Subject recruitment, experimental protocol, and data acquisition have been explained in detail previously (*Jones et al., 2010*). In brief, 10 subjects were instructed to fixate on a cross on a projection screen. Each trial lasted 3500 ms, and began with a 60 dB, 2 kHz tone delivered to both ears, simultaneously accompanied by the fixation cross changing into a visual word cue instructing the subject where to attend; either 'Hand' (attend-in condition), 'Foot' (attend-out condition) or 'Either' location. The tactile stimulus was delivered to the cued location at a randomized time between 1100 ms and 2100 ms after the cue onset. The stimulus was a single cycle of a 100 Hz sine wave (10 ms duration) generated by piezoelectric benders, as in the detection task. Stimuli were applied to the distal pads of the third digit of the left hand or first digit of the left foot, and PEST procedure (*Dai, 1995*; *Leek, 2001*) was employed before the task to estimate subject's initial detection threshold for both the hand and the foot. Both the auditory tone and the visual cue ceased after 2500 ms. Subjects then had 1000 ms to report detection or non-detection of the stimulus at the cued location, with button presses using the second and third digits of the right hand, respectively.

The task consisted of at least 5 cued detection runs, where each run consisted of 40 of each attention condition randomly intermixed (i.e. 120 trials per run), in one data collection session. Similar to the detection task, all analyses were limited to the last 100 perceived trials of attend-in and attend-out conditions for each subject. Note, limiting to perceived trials allowed us to dissect attention effects independent of detection performance.

## Mouse recordings

### Chronic extracellular electrophysiology in mice

All electrophysiology data was collected using the Open Ephys system. Continuous data was sampled at 30 kHZ, and off-line downsampled to 1000 Hz. To reject common noise shared across channels, such as muscle artifact, independent component analysis (ICA; https://research.ics.aalto.fi/ica/fastica/code/dlcode.shtml) was applied to the downsampled continuous data collected across all 64 channels throughout the entire duration of the session. Components resembling artifact were

manually chosen, and denoised signal was reconstructed with the exclusion of the rejected components. This reconstructed signal was used for all further analyses. As all electrodes were located in the barrel cortex and showed highly correlated LFP, one electrode was chosen for each mouse for all analyses.

## Detection task

### Animals

Two neurologically healthy male mice were used in this experiment (5 sessions from each mouse). Mice were 8–15 weeks at the time of surgery. Animals were individually housed with enrichment toys and maintained on a 12 hr reversed light-dark cycle. All experimental procedures and animal care protocols were approved by Brown University Institutional Animal Care and Use Committees and were in accordance with US National Institutes of Health guidelines.

### Surgical procedure

Naive mice were induced with isofluorane gas anesthesia (0.5–2% in oxygen 1 L/min) and secured in a stereotaxic apparatus. We injected slow-release buprenorphine subcutaneously (0.1 mg/kg; as an analgesic) and dexamethasone intraperitoneally (IP, 4 mg/kg; to prevent tissue swelling). Hair was removed from scalp with hair-removal cream, followed by scalp cleansing with iodine solution and alcohol. Then, skull was exposed by incision. After the skull was cleaned, muscle resection was performed on the left side. A titanium headpost was affixed to the skull with adhesive luting cement (C and B Metabond). Two small stainless-steel watch screws were implanted in the skull plates; one anterior to bregma, one on the right hemisphere. Next, a ~ 1.5 mm–diameter craniotomy was drilled over barrel cortex of the left hemisphere, and subsequently duratomy was performed. The guide tube array (8 by 2 arrangement of 33ga polyimide tubes; 2 mm by 0.5 mm) was centered at 1.25 mm posterior to bregma and 3.25 mm lateral to the midline and angled 45 degrees relative to midline. The drive body was angled 30 degrees relative to the perpendicular direction to compensate for the curvature of barrel cortex. Once the implant was stably positioned, C and B Metabond and dental acrylic (All for Dentist) was placed around its base to seal its place. A drop of surgical lubricant (Surgilube) prevented dental acrylic from contacting the cortical surface. Mice were given at least 3 days to recover before the start of water restriction.

### Trial structure and behavior control

The behavioral task setup was adapted from a previous study from our group (*Siegle et al., 2014*) with slight modifications. Each trial consisted of right-side (contralateral) vibrissae stimulation in the caudorostral direction, with 20 Hz deflections that lasted 500 ms (10 pulses with the same amplitude). The stimulus was delivered through piezoelectric benders (Noliac). Vibrissae on the right side were tied up with a suture loop fed through a glass capillary tube (0.8 mm outer diameter) attached to the piezoelectric bender. Most of the vibrissae were secured about 3 mm from the mystacial pad.

If the mouse licked within 700 ms relative to the onset of the stimulus, a drop of water was delivered, shortly followed by vacuum suction to remove any remaining water not consumed by the animal. Water delivery and vacuum suction was controlled by a solenoid valve (NResearch) connected with Tygon tubing. Mice received water through a plastic tube, which was positioned near the animal's mouth using a Noga arm. The water was delivered based on gravitational flow and the volume was controlled by the duration of valve opening (~100 ms), calibrated to give an ~3 µl per opening. Individual licks were detected using beam breaks of IR detector flanking the lick tube. Unlike the human detection task, there was no cue indicating the start of a trial.

All behavioral events including piezoelectric control, reward delivery, and lick measurements were monitored and controlled in Matlab and interfaced with a combination of Arduino and PCI DIO board (National Instruments, Austin, Texas).

### Behavioral training

Mice were water restricted for at least 7 days before start of training, during which time mice were acclimated to the head-fixed setup where mice could freely run on a fixed-axis styrofoam ball. Mice were given at least 1 ml per day, calibrated such that mice would not lose weight further than 80% of their original weight before water restriction.

Training began with reward-all session where vibrissae stimuli were paired with water delivery regardless of the animal's response. This allowed mice to establish an association between vibrissae deflection and reward, such that the mice learned to lick when they detected the stimulus in anticipation of the reward. Mice learned the association after about a week of reward-all training. After this period reward was only delivered on trials where the mouse licked within 700 ms of the stimulus onset (i.e. reward window).

The stimulus amplitude was varied on a trial by trial basis between 0 to maximal amplitude (about 1 mm deflection) in a randomized manner. Before the start of each session, the experimenter set the percentage of maximal trials and 0 amplitude (catch) trials. The rest of the trials were submaximal trials, where the stimulus amplitude was randomly drawn from a uniform distribution between 0 and maximal amplitude. Throughout training, percentage of maximal amplitude trials were gradually lowered to 10%, and percentage of catch trials were gradually increased to 25%.

Correct rejection trials were trials where the mouse did not lick during the reward window. On the contrary, false positives, where mice did lick during catch trials, led to a time out of 15 s. Further, exploratory licking during inter-trial intervals (ITI, 2–6 s) led to resetting (prolongation) of ITI, and this reset could happen up to 10 times. This prevented excessive impulsive licking.

If mouse did not consume enough water during the behavior session, supplementary water was given several hours after the conclusion of the session, such that mouse would have drank at least 1 ml of water each day. Mouse weight was monitored throughout the entire duration of training such that their weight would not fall below 80% of their original post-surgery weight. Mice performing the detection task was recorded over a period of ~3 months.

## Behavioral analysis

Only the behavior sessions that exhibited well-trained behavior were selected for analysis, based on the following criteria: (1) Stereotyped reaction time, as assessed by a clear peak in the reaction time histogram. (2) Goodness of fit for the Boltzmann distribution fit of psychometric performance over the entire session (mean of R-squared across the 10 sessions was 0.91, range was 0.77 ~ 0.99; mean of root mean squared error (RMSE) was 0.12, range 0.04 ~ 0.18).

Threshold level stimulus trials were chosen offline from periods within the <2.5 hr session where the mouse was engaged in the task. Even in well trained mice, there were periods where animals defaulted to non-optimal strategies, such as excessive impulsive licking or non-engagement from satiety or prolonged inattention. To address the issue of exploratory licking that was not in response to detection, hit trials were defined as trials where the animal not only licked within the 700 ms reward window but also did not lick the spout prestimulus up to −1000 ms. In addition, we filtered out the trials between two false positives if there were no misses or correct rejections in between, to exclude periods where the mouse was acting impulsively. To filter out periods where the mouse was not engaged in the task, we looked at trials with strong stimuli (as defined as stimulus at 80% detection, calculated from Boltzmann distribution fit of psychometric performance over the entire session). If three consecutive strong stimuli lead to non-detection and there were no detected trials in between, all trials in between were filtered out.

We chose stimulus amplitude-matched threshold level detected and non-detected trials for all analyses, with the exception of *Figure 1—figure supplement 1*. To do this, we binned all submaximal stimulus amplitude into 15 bins. Detected and non-detected trials were chosen from the high-performance periods to maximize trial count while matching the stimulus amplitude histogram. In all of the sessions analyzed, there were at least 100 stimulus amplitude-matched detected and non-detected trials each (mean 120 trials, range 106 ~ 160).

## Common data analysis procedures for human and animal recordings

### Evoked response

The evoked response to suprathreshold tactile stimulus was averaged first for each subject/session, then averaged across subjects/sessions. The source localized human MEG evoked response (*Figure 1—figure supplement 1Ai*) from a suprathreshold level brief tap to the contralateral third finger-tip is reproduced from (*Jones et al., 2007*), which is the average of 7 (out of 10) subjects in the human detection dataset analyzed in this paper. The mouse LFP evoked response from a 500 ms, 20

Hz deflection of contralateral vibrissae is depicted as the across-sessions average of the mean evoked response in maximal stimulus amplitude trials (*Figure 1—figure supplement 1Aii*).

## Spectral analysis

For all three tasks, we analyzed for each trial the −1000 to 0 ms window relative to the stimulus onset. The spectrograms of the spontaneous data were calculated for each prestimulus window by convolving the signals with a complex Morlet wavelet of the form:

$$w(t, f_0) = A \exp\left(-\frac{t^2}{2\sigma_t^2}\right) \exp(2i\pi f_0 t)$$

for each frequency of interest $f_0$, where $\sigma = m/2\pi f_0$ and $i$ is the imaginary unit. For both the human datasets, the spectrogram was calculated from 1 to 60 Hz, whereas for the mouse dataset, the spectrogram was calculated from 1 to 100 Hz. The normalization factor was $A = 1/\sigma_t\sqrt{2\pi}$.

Consistent with prior publications from our group (*Sherman et al., 2016*), we chose 7 for constant $m$ (number of Morlet wavelet cycles). Time-frequency representations of power (TFR) were calculated as the squared magnitude of the complex wavelet-convolved data.

In all analyses involving the Morlet spectrogram, the TFR was normalized by the median power value for each frequency. This median was calculated from all power values, at each frequency, in the −1000 to 0 ms prestimulus TFR concatenated across trials. Normalized TFR values are calculated in factors of median (FOM) for each frequency, separately for each subject/session.

### Trial mean prestimulus beta power

For trial mean prestimulus power, the normalized prestimulus TFR was averaged across time (−1000 to 0 ms window) and frequency band (15 to 29 Hz, inclusive), in each trial.

## Defining beta events and features

### Power cutoff for beta events

In all Figures except *Figures 11* and *12*, beta events were defined as local maxima in the trial-by-trial TFR matrix for which the frequency value at the maxima fell within the beta band (15–29 Hz) and the power exceeded a set cutoff. Local maxima were found using the Matlab function 'imregionalmax'. Custom software for identifying beta events and calculating event features is written in Matlab and available at https://github.com/hs13/BetaEvents (*Shin, 2017*; copy available at https://github.com/elifesciences-publications/BetaEvents). 'Non-overlapping beta event' definition and analysis and Dynamic amplitude modulation versus bursty generator analysis below for beta event definitions in *Figures 11* and *12*, respectively.

To choose the power cutoff that best captures variability in trial mean prestimulus power, we calculated the percent area in −1000 to 0 ms prestimulus beta-band spectrogram above power cutoff. That is, for the −1000 to 0 ms prestimulus beta-band TFR matrix in each trial, we quantified the percentage of matrix elements (i.e. pixels in spectrogram) that has power above cutoff. For all analyses shown, except *Figure 4*, *Figure 5—figure supplement 1*, *Figure 6—figure supplement 2*, *Figure 7—figure supplement 1*, and *Figure 8—figure supplement 1*, the power cutoff was set to be 6X the median power.

### Beta event number

Beta event number was calculated as the number of beta events in the −1000 to 0 ms window for each trial.

### Beta event power

Beta event power is defined as the normalized TFR value (unit of FOM) at the local maximum that defines each event. The trial mean event power was defined as the power of all events averaged in the −1000 to 0 ms prestimulus period. For all analyses of trial mean event power, only trials that had at least one event were considered.

## Beta event duration and frequency span

Beta event duration and frequency span were defined as full-width-at-half-maximum from the beta event maximima in the time and frequency domain, respectively. Edge cases in the time domain were handled in the following way: for maxima happening near the edge time points (−1000 ms and 0 ms), if the power did not fall below the half-maximum at the edge, event duration was calculated by doubling the half-width of the side that was not cut by the relevant edge. The same method was used to handle edge cases for frequency span; if the power did not fall below half-maximum at either the lower boundary (1 Hz) or at the upper boundary (60 Hz for human, 100 Hz for mice), frequency span was calculated by doubling the half-width of the side that was not cut by the boundary.

The trial mean event duration and frequency span were defined as the averaged values in the −1000 to 0 ms prestimulus window. For all analyses of trial mean beta event duration and frequency span, only the trials that had at least one event was considered.

## 'Most recent beta event' timing and other features

The 'most recent event' for each trial was defined as the event that happened closest to the stimulus onset in the −1000 to 0 ms prestimulus window. 'Most recent event' timing was defined as the timing of the maximum of that event. Other features of the 'most recent event' (power, duration and frequency span) were calculated as defined above. With the exception of *Figure 8i*, only the trials that had at least one event was considered for all analyses pertaining to the 'most recent event'.

## Details of data analyses
### Temporal evolution of beta power

In *Figure 1i*, mean prestimulus beta power as a function of time relative to stimulus onset was calculated by first averaging the prestimulus TFR in each trial across the frequency band (15 to 29 Hz, inclusive). The resulting time-dependent beta-band power was then averaged for each behavioral condition, for each subject/session. At each timepoint, a left-tailed Wilcoxon signed rank test was applied across subjects/sessions to test beta power was higher on non-detected trials. We only show −900 to −100 ms, as the Morlet wavelet convolution introduces edge effects.

### Time histogram of event rate

In *Figure 7i*, the probability time histogram of beta event occurrence in the 1 s prestimulus window was calculated in 50 ms windows sliding in 1 ms steps, for each behavioral condition and each subject/session. For each window, a left-tailed Wilcoxon signed rank test was applied across subjects/sessions to test whether an event was more likely to happen within that window on non-detected trials. We only show −900 to −100 ms, as the Morlet wavelet convolution introduces edge effects.

### 'Most recent event' probability time histogram

*Figure 8i* depicts the probability of the 'most recent event' happening at a certain time bin, given a detected/non detected trial, for each subject/session. Therefore, we included all detected/non detected trials (at perceptual threshold) for this analysis. The probability histogram was calculated in 50 ms windows sliding in 1 ms steps. For each window, a left-tailed Wilcoxon signed rank test was applied across subjects/sessions to test whether the 'most recent event' was more likely to happen within that window on non-detected trials. We only show −900 to −100 ms, as the Morlet wavelet convolution introduces edge effects.

### Trial percentile and hit rate/attend in rate relationship

*Figure 1ii*, *7ii* and *8ii* were intended for the visualization of the relationship between each variable under consideration (e.g. prestimulus trial mean power) and the hit rate/attend in rate. First, all trials were sorted in increasing order of the variable under consideration. Then, the detection/attention rate was calculated in boxcar windows of 21 trials, sliding in 1-trial steps; where the first bin was 21 bottom-ranking trials and the last bin was 21 top-ranking trials. If multiple trials had the same value (e.g. multiple trials with one event), the sorted trial order was shuffled to prevent artificial correlation. Detection/attention rate was then normalized as percent change from mean (PCM).

The human data was analyzed across 100 trials per behavioral condition in a single session, as described above. The mouse data had an uneven number of trials across sessions. Therefore, we

used the Matlab 'interp1' function to resample 100 bins of the detection/attention rate. The resampled detection/attention rates were averaged across sessions/sessions, and the standard error of the mean were calculated.

## Pooled average analysis

For each pooled average analysis reported, all prestimulus data under consideration for a given behavioral condition were averaged. Only trials with at least one event were considered, with the exception of trial mean prestimulus beta power and event number. Right-tailed paired t-test was used to test whether the pooled averages of non-detected trials was significant higher than detected (attend-out was significantly higher than attend-in) across subject/sessions.

## Effect size (Cohen's d)

The effect size quantifies the difference between behavioral conditions. For the human and the mouse detection datasets, we quantified the (miss – hit) difference; for the human attention dataset, we quantified the (attend out – attend in) difference. We employed the Cohen's d measure, which was defined as follows:

$$d = \frac{\bar{x}_1 - \bar{x}_2}{\sqrt{\frac{(n_1-1)s_1^2+(n_2-1)s_2^2}{n_1+n_2-2}}}$$

where $\bar{x}_1$ and $\bar{x}_2$ denote the mean of the two populations (e.g. miss versus hit or attend-out versus attend-in) being compared, an $n_1$ and $n_2$ denotes the number of sessions in each population, and $s_1$ and $s_2$ are the standard deviation of each population.

## Detect Probability/Attend Probability analysis.

Detect probability (DP)/attend probability (AP) was calculated as the area under a receiver operating characteristic (ROC) curve, where the ROC curve was defined by applying a binary classification of each variable under consideration (e.g. trial mean prestimulus beta power) to the behavioral condition. The area under the ROC curve analysis was performed with the Matlab function 'perfcurve'. For all analyses, only trials with at least one event were considered, with the exception of trial mean prestimulus beta power and event number.

DP/AP of 0.5 indicates that the beta event feature under consideration cannot dissociate between the behavioral condition. DP/AP under 0.5 signifies that beta event feature under consideration is significantly predictive of non-detected/attend out trials. One-sample left-tailed Wilcoxon signed-rank test was used to determine whether the median of the DP/AP distribution was significantly less than 0.5. To quantify the percentage of significant subjects/sessions, a 95% confidence interval was determined individually for each subject/session by bootstrapping 1000 times. If the upper boundary of the 95% confidence interval was below 0.5, the beta event feature under consideration was considered significantly predictive of non-detected/attend out trials for that subject/session.

## Correlation with prestimulus trial mean power

In *Figure 6*, Pearson's correlation with trial mean prestimulus beta power was calculated on a trial-by-trial basis for trial summary of each beta event feature (i.e. beta event number per trial, trial mean beta event power, duration and frequency span). To determine which beta event feature best correlated with trial mean prestimulus beta power, we first ran a Friedman test to assess significant differences in correlation coefficient distributions. When Friedman test returned significance, we ran a post-hoc Wilcoxon signed-rank test, corrected for multiple comparisons using a Holm-Bonferroni method.

## Power cutoff variation analysis

Correlation with trial mean prestimulus beta power (*Figure 6—figure supplement 2*) and % subjects with DP/AP significantly over cutoff (*Figure 7—figure supplement 1*, *Figure 8—figure supplement 1*) were calculated for a wide range of thresholds, including: 0.25, 0.5, 1 ~ 16 (integer values). For each dataset, feature, and cutoff, we tested whether the DP/AP distribution across subjects/sessions was significantly less than 0.5 (one-sample left-tailed Wilcoxon signed-rank test).

## Optimal event number criterion analysis

In *Figure 7–figure supplement 2ii*, we considered a binary classifier that categorized trials as non-detected/attend out when (event number ≥criterion). The optimal criterion for each subject/session was the criterion at which the binary classifier achieved the highest performance, as measured by (True Positive Rate – False Positive Rate).

## Histogram matching for event number and 'most recent event' timing

Histogram matching was achieved by random trial trimming process for *Figure 9*.

The event number matching was done as follows:

(i) In each subject/session, we iterated through event numbers from 1 to max number of events per trial.

(ii) In each event number bin (bin size = 1), we looked at trials with the given event number; if there were more detected (N = *h* trials) than non-detected (N = *m* trials, *h* < *m*) trials in that event number bin, we randomly selected *h* trials out of non-detected trials, and vice versa if *h* ≥ *m*.

The 'most recent event' timing matching was done as follows:

(i) In each subject/session, we iterated through 'most recent event' timing bins, where bin size was 20 ms, and the first bin started at −1000 ms and the last bin ended at 0 ms.

(ii) In 'most recent event' timing bin, we looked at trials that had the 'most recent event' in that time bin; if there were more detected (N = *h* trials) than non-detected (N = *m* trials, *h* < *m*) trials in that time bin, we randomly selected *h* trials out of non-detected trials, and vice versa if *h* ≥ *m*.

## Inter-Event Interval (IEI)

In *Figure 10i*, the IEI was calculated as the time difference between two consecutive events, in trials that had two or more events. Note that because of the 1000 ms window limit, IEI exceeding the window size could not be captured. The theoretical lower limit (resolution) of IEI was determined by the 7-cycle Morlet wavelet we used to generate the TFR matrices. All IEI values were pooled across trials within subjects/sessions, and the probability histograms were calculated in 15 ms bins.

## Coefficient of variation ($CV^2$) and Fano Factor (FF)

In *Figure 10ii*, Coefficient of variation of the inter-event intervals was defined as follows:

$$CV^2 = \frac{Var[\boldsymbol{IEI}]}{E[\boldsymbol{IEI}]^2}$$

where $Var[\boldsymbol{IEI}]$ is the variance, and $E[\boldsymbol{IEI}]$ is the mean.

Fano factor was used to quantify the trial-to-trial variability of event number per trial (EpT), and was defined in the following way:

$$FF = \frac{Var[\boldsymbol{EpT}]}{E[\boldsymbol{EpT}]} \qquad \text{[EpT]}$$

Renewal processes are defined as point processes with independent and identically distributed waiting times. Renewal processes are characterized as having equal $CV^2$ and FF values for appropriately long trial lengths. Our results are limited by the fact that all analysis is restricted to −1000 to 0 ms prestimulus.

## 'Non-overlapping beta event' definition and its features

In *Figure 11*, each continuous supra-cutoff temporal region was defined to be a 'non-overlapping event', using the 6X median power cutoff.

With this definition, multiple local maxima in the beta band (i.e. the original definition of beta events) could occur within one 'non-overlapping beta event'. Therefore, the number of local maxima per 'non-overlapping beta event' was quantified, for each behavioral condition (*Figure 11i*). The 'non-overlapping beta event' power, duration and frequency span were defined in the same way as the original definition of beta events: Power was defined as the maximum power within the supra-cutoff region; and the 'non-overlapping beta event' duration and frequency span were defined as the full-width at half-maximum in the time and frequency domain, respectively (edge cases were

handled the same way as section Beta event duration and frequency span). The 'non-overlapping beta event' timing was defined as the timing of the maximal power within each supra-cutoff region.

### 'Non-overlapping beta event' IEI calculation and jitter method

In *Figure 11ii*, the 'non-overlapping beta event' IEI was defined as the difference in timing of each successive 'non-overlapping beta events'. The jittered 'non-overlapping beta event' IEI distribution was generated by randomly jittering the timing of 'non-overlapping beta events', within the 1 s prestimulus window of the events' respective trials. The limiting conditions for the jittering was that (i) the independent events stayed non-overlapping; and (ii) the relative order in which the events occurred within a trial was preserved. This jitter process was repeated 1000 times and averaged for each subject/session.

### Dynamic amplitude modulation versus bursty generator analysis

For the analysis in *Figure 12*, we first band-pass filtered the raw 1 s prestimulus data for each trial separately, in the beta band (15–29 Hz) using a 150 ms long Hamming window based FIR filter (eegfilt function in EEGlab https://sccn.ucsd.edu/eeglab/index.php). We then determined beta phase and amplitude by Hilbert-transformation of the band-pass filtered data. We chose the Hilbert-amplitude cutoff to be 2X median of beta band-passed Hilbert amplitude cutoff. This cuttoff approximately matched the 6X median of Morlet spectral power cutoff used to define beta events throughout the paper, in that the proportion of timepoints above the cutoff in the 1 s prestimulus window was approximately 10% across subjects/sessions in both cases.

We then determined the characteristic beta period for each subject/session as the inverse of the characteristic beta frequency. In turn, the characteristic beta frequency for each subject/session was defined as the median of the instantaneous frequency values in the Hilbert-transformation of the beta band-passed data. This characteristic beta period for each subject/session was used to calculate the time lags' modulo of the beta period, for time lags above and below the amplitude cutoff. Modulo operation calculates the remainder after division; for example, time lag modulo the beta period of 0 would mean that the time lag was an integer multiple of the beta period. To compare across subjects/sessions, we normalized the time lag modulo of the beta period values by the characteristic beta period, such that the normalized values would vary between $-0.5$ and $0.5$.

## Acknowledgements

We thank Christopher Black for help with editing figures on Adobe Illustrator and Dr. Manuel Gomez-Ramirez for comments on the manuscript. This work was supported by the US National Institutes of Mental Health (R01MH106174; SRJ); the National Science Foundation Collaborative Research in Computational Neuroscience (NSF CRCNS-1131850); the Department of Veterans Affairs, Veterans Health Administration, Office of Research and Development, Rehabilitation Research and Development Service, Project N9228-C (SRJ); US National Institute of Neurological Disorders and Stroke (R01NS045130; CIM); and fellowships from Fulbright and the Brown Institute for Brain Sciences to HS.

## Additional information

### Funding

| Funder | Grant reference number | Author |
| --- | --- | --- |
| Brown Institute for Brain Science | | Hyeyoung Shin |
| Fulbright Association | | Hyeyoung Shin |
| National Institute of Neurological Disorders and Stroke | R01NS045130 | Christopher I Moore |
| National Institute of Mental Health | R01MH106174 | Stephanie R Jones |

| | | |
|---|---|---|
| U.S. Department of Veterans Affairs | Veterans Health Administration, Office of Research and Development, Rehabilitation Research and Development Service; N9228-C | Stephanie R Jones |
| National Science Foundation | Collaborative Research in Computational Neuroscience NSF CRCNS-1131850 | Stephanie R Jones |
| National Institute of Biomedical Imaging and Bioengineering | RO1EB022889 | Stephanie R Jones |

The funders had no role in study design, data collection and interpretation, or the decision to submit the work for publication.

## Author contributions

Hyeyoung Shin, Conceptualization, Data curation, Software, Formal analysis, Funding acquisition, Validation, Investigation, Visualization, Methodology, Writing—original draft, Writing—review and editing; Robert Law, Conceptualization, Formal analysis, Validation, Investigation, Methodology, Writing—review and editing; Shawn Tsutsui, Conceptualization, Formal analysis, Investigation, Methodology; Christopher I Moore, Conceptualization, Resources, Supervision, Funding acquisition, Investigation, Writing—original draft, Project administration, Writing—review and editing; Stephanie R Jones, Conceptualization, Resources, Data curation, Formal analysis, Supervision, Funding acquisition, Investigation, Methodology, Writing—original draft, Project administration, Writing—review and editing

## Author ORCIDs

Hyeyoung Shin http://orcid.org/0000-0002-7587-8577
Shawn Tsutsui http://orcid.org/0000-0003-3805-1519
Stephanie R Jones http://orcid.org/0000-0001-6760-5301

## Ethics

Human subjects: All MEG experimental protocols were approved by the Massachusetts General Hospital Internal Review Board, and each subject gave informed consent before data acquisition.
Animal experimentation: All experimental procedures and animal care protocols were approved by Brown University Institutional Animal Care and Use Committees and were in accordance with US National Institutes of Health guidelines. All surgery was performed under isofluorane anesthesia, and every effort was made to minimize suffering.

## Decision letter and Author response

Decision letter https://doi.org/10.7554/eLife.29086.024
Author response https://doi.org/10.7554/eLife.29086.025

# Additional files

## Supplementary files

• Transparent reporting form
DOI: https://doi.org/10.7554/eLife.29086.021

## Major datasets

The following dataset was generated:

| Author(s) | Year | Dataset title | Dataset URL | Database, license, and accessibility information |
|---|---|---|---|---|
| Shin H, Jones SR, Moore CI | 2017 | Data from: The rate of transient beta frequency events predicts behavior across tasks and species | http://dx.doi.org/10.5061/dryad.pn931 | Available at Dryad Digital Repository under a CC0 Public Domain Dedication |

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
