## [Decision Letter]

Thank you for submitting your article "The rate of transient beta frequency events predicts impaired function across tasks and species" for consideration by *eLife*. Your article has been reviewed by two peer reviewers, and the evaluation has been overseen by a Reviewing Editor and Sabine Kastner as the Senior Editor. The reviewers have opted to remain anonymous.

The reviewers have discussed the reviews with one another and the Reviewing Editor has drafted this decision to help you prepare a revised submission.

Summary:

This study shows that beta band signals appear as short beta events using human MEG and mouse LFP in barrel cortex, and investigates how apparent change in beta band power and somatosensory detection/attention relate to the event features (i.e. event rate, duration, spectral width, amplitude and timing) within individual trials of tasks. Results show that the event rate of beta consistently correlates with sensory outcome and beta power, and particularly recent events are the strongest predictors.

All reviewers agree with the novelty and potential importance of findings documented in the manuscript. Though well written, there are many issues that need clarification, and there are several concerns that should be addressed.

Essential revisions:

1) The study's conclusions are not constrained as long as it is not documented from which neural populations beta events are measured. Enhanced transient beta event rates in a neural population that is not relevant for the near threshold sensory detection would have a fundamentally different functional meaning than a reduced event rate in a neural population that is essentially encoding task relevant information. But the current manuscript version does not disambiguate from which neural populations these events were measured. It is unclear from which sensor level or deep sources the beta events in the human datasets originate, nor is it reported whether the beta events in barrel cortex stem from neural recording sites that were overlapping the stimulation site or not, or from sites with evoked responses to the stimulation.

Since there are more than 300 sensors measured in the human datasets it might be safe to assume that only a very small portion of them are measuring activity from neural circuits necessary for efficient preparation of the detection response. The enhanced beta event rate may thus be dominated by activity from neural populations that interfere with the task, including from sensory or language related areas, from motor related areas or from prefrontal areas where beta activity may signify activation rather than inhibition. In this situation, it seems necessary that the authors report whether event rates changes were differentially observed in those recording/measurement sites that are apparently task modulated as opposed to those not task modulated.

Thus, the manuscript should show information about locations of MEG sources and LFP recordings. If signals were from the hand area of somatosensory cortex and barrel cortex, then somatosensory responses to finger taps and whisker deflections are expected. Additionally, those sensory neuronal responses could be susceptible to prior beta events, similarly to behaviorally-indicated sensory detection. In human attention tasks, it is also possible to see beta band of dipole in foot area acting differently from beta band in hand area depending on the cue in each trial.

In a similar vein, it would be helpful to know whether the beta event rate enhancement was more a frontal / motor preparation related effect, or a sensory association cortex effect.

2) While the unconventional analysis approach to beta events is timely and interesting, it comes with the risk of "relabeling" something we already know with novel terminology.

The starting point of the manuscript is the hypothesis that neural activity in the beta frequency band is best described as bursts. While there is certainly some evidence that supports this assumption, it would be good if the authors could also provide some key evidence for this in the current manuscript. Whereas the (implicit) alternative model used by the authors appears to be a model with stationary oscillatory amplitude, it may be more useful here to consider an alternative oscillation-based model with dynamic amplitude modulations.

There are at least two further analyses that may adjudicate burst vs. dynamic amplitude accounts. First, the authors could provide histograms of Hilbert-transformed amplitude envelope data. If there are two regimes ([no-bursts + noise] and [bursts+noise]), then this should result in amplitude distributions (histograms) with bimodal features (in the clearest case: a separate 'bump' at the participant-specific burst amplitude). Second, the authors may look at the degree of phase-preservation between successive time points as well as between successive bursts. Under a dynamic amplitude account, one would expect consistent phase relations between successive time points / successive "bursts". In contrast, the bursting account, as far as I understand it, would predict phase independence between beta "oscillations" at successive time points and bursts.

3) A related issue regards the concern that epochs with prominent (and largely sustained) beta oscillations may be wrongfully classified as epochs with high burst rate. Noise (as well as dynamic amplitude fluctuations) may tip the beta estimates above and below threshold, resulting in multiple "bursts" (e.g., high burst rate) in the authors' analysis. This appears particularly a concern given that most inter-burst-intervals (Figure 9) indeed appear to occur within the range of few millisecond only. Have the authors considered imposing a minimum interval between successive bursts (and/or a minimal amplitude relaxation between successive bursts)? Would the same results be obtained?

Figure 5 and Figure 9 (duration ~ 100 ms and intervals of <50 ms) together suggest that many bursts occur as doublets of about 250 ms, or triplets. Is that so? If yes, there should be more to characterize, like the timing of doublets. It may be noted that 250 ms or less is about the duration of the prestimulus period described as when beta events are effective in modulating sensory detection.

In addition, Figure 9 shows the interval distributions do not differ between behavioral outcomes. It looks inconsistent with that more beta events occur in miss trials than hit trials.

4) For comparison, it appears key to also include outcomes of regular power analyses when relating neural activity to behavioral performance, as in Figure 7 (for the 1s pre-stimulus epoch) and Figure 8 (power as a function of time). Are events features (rate) really better predictors of performance than conventional power estimates?

5) The authors limit analysis to the beta frequency range, but in the discussion mention alpha and gamma bands. One might expect these frequency ranges to be modulated as well at least in a fraction of sensors and recording sites. Why did the authors not add a brief summary analysis on whether similar findings as the beta event rate changes might be obtained with alpha and gamma ? In that way one immediately can discern whether the findings have clear frequency specificity, or whether similar effects can be expected in other bands with clear power spectral peaks. Either of these results would be highly informative.

If the authors find this analysis would exceed the scope of the study, the discussion on alpha and gamma should be shortened and instead a discussion of the possibly different functional roles of the beta frequency (sub)ranges is included to discern better why the analysis concentrates on this band and whether the authors suggest that the band represents some homogeneous underlying sources (or not).

Also, there is evidence that rolandic alpha oscillations also predict somatosensory performance and are modulated by somatosensory attention. Have the authors considered extending their analyses to this frequency range?

6) What was the proportion of trials in each dataset with at least one event (the inclusion criteria) in correct and error trials? Without this information, it is difficult to discern how important the beta events would be to predict trial-by-trial performance. A discussion of the overall effect size considering this number would be very interesting to readers – this is particularly relevant with regard to the stimulation approaches that would want to utilize event like stimulation protocols.

Do the curves in Figure 8 derived from all trials or a fraction of trials that had beta events? Event rate can be derived in both ways. Blue and red curves in the figure should be derived from all detected and non-detected trials respectively.

Which trials were used to derive the threshold in Figure 4? All Hit, Miss, FA, and catch trials?

Would the strength of the correlations look very different when including these zero event trials as zero's and using Spearman instead of Pearson correlations?

---

## [Author Response]

Essential revisions:1) The study's conclusions are not constrained as long as it is not documented from which neural populations beta events are measured. Enhanced transient beta event rates in a neural population that is not relevant for the near threshold sensory detection would have a fundamentally different functional meaning than a reduced event rate in a neural population that is essentially encoding task relevant information. But the current manuscript version does not disambiguate from which neural populations these events were measured. It is unclear from which sensor level or deep sources the beta events in the human datasets originate, nor is it reported whether the beta events in barrel cortex stem from neural recording sites that were overlapping the stimulation site or not, or from sites with evoked responses to the stimulation.Since there are more than 300 sensors measured in the human datasets it might be safe to assume that only a very small portion of them are measuring activity from neural circuits necessary for efficient preparation of the detection response. The enhanced beta event rate may thus be dominated by activity from neural populations that interfere with the task, including from sensory or language related areas, from motor related areas or from prefrontal areas where beta activity may signify activation rather than inhibition. In this situation, it seems necessary that the authors report whether event rates changes were differentially observed in those recording/measurement sites that are apparently task modulated as opposed to those not task modulated.Thus, the manuscript should show information about locations of MEG sources and LFP recordings. If signals were from the hand area of somatosensory cortex and barrel cortex, then somatosensory responses to finger taps and whisker deflections are expected.

We thank the reviewer for pointing out this lack of clarity. The data from both the human detection and attention datasets represents activity from a dipole signal source localized to the hand area of SI. This was detailed in our prior publications (Jones et al., 2010, Jones et al., 2007) but not clearly stated in the current paper. In the mouse dataset, the chronic electrode implant (flexDrive, Voigts et al., 2013) was positioned over barrel cortex during a stereotaxic surgery and recording sites had a clear evoked response in the local field potentials in response to vibrissae deflections.

We have clarified these issues in several ways.

1) We have added a new figure that shows the tactile evoked response in the area recorded from the human and mouse data sets: see Figure 1—figure supplement 1. (Figure 1—figure supplement 1 is discussed in our response to reviewers’ comment (1)).

2) We have added the details of the localization methods in “Mean prestimulus beta power is higher on non-detected and attend-out trials” section of Results, and Figure 1 legend.

3) We state details of the human source localization in the Materials and methods section:

“Dipole activity from a suprathreshold level stimulus to the third digit of the right hand was localized to the postcentral gyrus in the contralateral hand area of SI using least-squares fit inverse methods and individualized structural MRIs, or a spherically symmetric conductor model of the head (Hamalainen & Sarvas 1989). All analyses presented in this paper are derived from the forward solution from this source-localized region of interest.”

We state details of the mouse electrode implant in the Materials and methods section:

“The guide tube array (8 by 2 arrangement of 33ga polyimide tubes; 2 mm by 0.5 mm) was centered at 1.25 mm posterior to bregma and 3.25 mm lateral to the midline and angled 45 degrees relative to midline. The drive body was angled 30 degrees relative to the perpendicular direction to compensate for the curvature of barrel cortex.”

Additionally, those sensory neuronal responses could be susceptible to prior beta events, similarly to behaviorally-indicated sensory detection. In human attention tasks, it is also possible to see beta band of dipole in foot area acting differently from beta band in hand area depending on the cue in each trial.In a similar vein, it would be helpful to know whether the beta event rate enhancement was more a frontal / motor preparation related effect, or a sensory association cortex effect.

We hope that we have clarified the signal localization methods in each of our data sets.

We agree with the reviewers that it would be very interesting to investigate the event-like characteristics of beta oscillations in other areas of the brain. However, given that the goal of the current paper was to inspect what features of transient beta “events” contribute to beta’s impact on function, we limited our analysis to the primary somatosensory area, where we have previously reported that beta robustly affects function (Jones et al., 2010, Jones et al., 2009). In addition, the striking homology of beta dynamics between the human datasets and the mouse dataset is a main thesis of our paper, and given that the mouse recordings were only conducted in the primary somatosensory areas, we would not be able to draw similar parallels for other regions of the brain. Therefore, we feel that it is beyond the scope of our paper, which already presents analyses in two tasks and two species, to characterize beta dynamics in other brain regions. Other studies recording LFPs in monkeys have studied the transient properties of beta oscillations and their relevance to function in other brain areas, including motor cortex (Lundqvist et al., 2016, Rule et al., 2017) and basal ganglia (Feingold et al., 2015). These papers are discussed in our Introduction and Discussion sections.

Of further note, the postcentral gyrus in the hand area of SI is particularly suitable for applying the inverse solution to extract source-localized data using MEG, due to its dendritic alignment being tangential to the MEG sensors. In prior analyses, we failed to reconstruct consistent dipoles from the foot localization data using the standard equivalent current dipoles localization techniques described above; therefore, we only analyzed and presented activity of the hand area. This fact is detailed in a prior report (Jones et al., 2010).

2) While the unconventional analysis approach to beta events is timely and interesting, it comes with the risk of "relabeling" something we already know with novel terminology.The starting point of the manuscript is the hypothesis that neural activity in the beta frequency band is best described as bursts. While there is certainly some evidence that supports this assumption, it would be good if the authors could also provide some key evidence for this in the current manuscript. Whereas the (implicit) alternative model used by the authors appears to be a model with stationary oscillatory amplitude, it may be more useful here to consider an alternative oscillation-based model with dynamic amplitude modulations.There are at least two further analyses that may adjudicate burst vs. dynamic amplitude accounts. First, the authors could provide histograms of Hilbert-transformed amplitude envelope data. If there are two regimes ([no-bursts + noise] and [bursts+noise]), then this should result in amplitude distributions (histograms) with bimodal features (in the clearest case: a separate 'bump' at the participant-specific burst amplitude). Second, the authors may look at the degree of phase-preservation between successive time points as well as between successive bursts. Under a dynamic amplitude account, one would expect consistent phase relations between successive time points / successive "bursts". In contrast, the bursting account, as far as I understand it, would predict phase independence between beta "oscillations" at successive time points and bursts.

This comment is important, and we conducted substantial new analyses to address it. As detailed below, we performed the suggested additional analyses and found they provide further evidence that SI beta rhythms are transient and “bursty” rather than rhythmic. This new result is included as new Figure 12.

Before we dive into our new analyses, we would first like to emphasize that in our view, the event-like properties of beta is most notable in its stereotyped waveform when *preselected* for high beta power, as extensively characterized in a prior publication from our group (Sherman et al., 2016). In good agreement with this idea, in the present paper we found that high power beta events (above 6X median power cutoff) indeed have stereotyped spectral features, where the duration and frequency span have less variability (Figure 5—figure supplement 1). Importantly, we further found that such high power regions explain most of the trial-by-trial fluctuations in beta power (Figure 4), suggesting that these high-power beta events may dominate beta’s impact on function. Given this emphasis on the stereotypy of the waveform (Sherman et al., 2016) and resultant spectral features (Figure 5—figure supplement 1), bimodality in spectral power is not a prerequisite for “burstiness” of beta events. To make this view more explicit, we added a discussion of this topic in the Discussion section of our paper.

Indeed, the histograms of Hilbert-transformed amplitude envelope analysis (as directly suggested by the reviewer) did not show a consistent pattern of bimodality in beta power (Author response image 1). Although there are hints of bimodality in some subjects / sessions, these distributions overall do not provide compelling evidence that the distribution is bimodal (see also inset in Figure 4ii for histogram of beta local maxima power).

**Author response image 1. respfig1:** Hilbert-transformed amplitude histograms do not consistently show bimodality. Hilbert-transformed amplitude histograms of band-pass (14~29Hz) filtered data for each subject / session in A. Human Detection, B. Mouse Detection, C. Human Attention datasets. The amplitude values were taken from all time points in the 1 second prestimulus window, and normalized as factors of median. The bandpass filtering was done with an FIR filter, using the eegfilt function in EEGlab (https://sccn.ucsd.edu/eeglab/index.php).

Per the second analysis suggested by the reviewer, we made explicit comparisons between two mechanisms that could underlie transient surges in beta power in the spectrogram: 1) the Dynamic Amplitude Modulation mechanism; and, 2) the Bursty Generator mechanism (see new Figure 12 schematic). The Dynamic Amplitude Modulation hypothesis predicts that an ongoing continuous rhythm exists, whose amplitude is dynamically modulated. Under this hypothesis, the rhythmicity of the beta component of the signal would be independent of its amplitude (i.e. spectral power). In contrast, the Bursty Generator hypothesis assumes that the generator that triggers the beta waveform also causes the transient surge in beta amplitude. Under this hypothesis, beta rhythmicity is only present when this generator triggers the system; therefore, rhythmicity is restricted to high beta power regions of the data. The contrasting predictions of these two competing hypotheses can be tested by choosing an amplitude (power) cutoff, and comparing the rhythmicity in regions above and below the cutoff.

The intuition behind this analysis is shown in Figure 12: As diagrammed, we assessed whether or not rhythmicity was maintained by comparing the time lag between first and last peaks in the oscillation within a high power region, or last and first peaks in consecutive regions with a low power region in between. Specifically, we band-passed the data and performed a Hilbert transform to calculate the time lag modulo the beta period for these regions. These values were plotted as functions of the normalized modulo (remainder after division) that varied between -0.5 and 0.5, where 0 corresponds to integer multiples of the beta period. The “time lag above cutoff” (red periods in Figure 12) were calculated as the time between the first (orange dot) and last (brown dot) 0° phase time points in each supra-cutoff region, where 0° phase denotes the peak points in the oscillation (marked with grey / orange / brown dots in Figure 12 schematic). The “time lag below cutoff” was calculated as the time between the last 0° phase timepoint in the preceding supra-cutoff region and the first 0° phase timepoint in the succeeding supra-cutoff region. With this definition, we would expect the time lags to be integer multiples of the beta period in rhythmic regions, i.e. we would expect the modulo of the beta period to be around 0.

Under the Dynamic Amplitude Modulation hypothesis, where a sustained oscillation is being modulated in amplitude, we would expect both the time lags above (red) and below (blue) the cutoff to be rhythmic, hence the time lag modulo the beta period distributions should both be centered at 0 (lower panels in Figure 12Ai). In contrast, the Bursty Generator hypothesis would predict that supra-cutoff regions would be rhythmic, whereas sub-cutoff regions are not; therefore, the time lag modulo the beta period distributions would be centered at 0 for time lags above cutoff (red), whereas the distribution would be flat for time lags below cutoff (blue) (lower panels in Figure 12Aii).

We chose the Hilbert-amplitude cutoff (cyan horizontal line in Figure 12) such that it was comparable to the 6X median power cutoff in the Morlet spectrogram analysis, in that there was a similar percentage of time points above the cutoff in the beta band (~10%). We determined the characteristic beta period for each subject / session as the inverse of the characteristic beta frequency. The characteristic beta frequency in turn was defined as the median of the instantaneous frequency values in the Hilbert-transformation of the beta band-passed data. This characteristic beta period was used to calculate the time lags’ modulo of the beta period and to normalize to range between -0.5 and 0.5.

To account for the possibility that rhythmicity may decay over time, we first performed histogram matching of time lags above and below cutoff such that the number of beta cycles in the time lags being compared was matched. The modulo probability histograms for the matched time lags above and below the amplitude cutoff are plotted in Figure 12 for each dataset (i-iii). These probability histograms show that time lags above cutoff are integer multiples of the beta period (i.e. distributions are centered at 0), in direct contrast to time lags below cutoff that are not (i.e. distributions are flat).

Overall, these results strongly support the prediction that high power regions are rhythmic but low power regions are not, providing evidence for the Bursty Generator hypothesis over the Dynamic Amplitude Modulation hypothesis.

We added these results and new Figure 12 in a new section of the Results subtitled, “BETA events are generated by a bursty mechanism, rather than by dynamic amplitude modulation of sustained beta oscillation”. See also Materials and methods for details.

3) A related issue regards the concern that epochs with prominent (and largely sustained) beta oscillations may be wrongfully classified as epochs with high burst rate. Noise (as well as dynamic amplitude fluctuations) may tip the beta estimates above and below threshold, resulting in multiple "bursts" (e.g., high burst rate) in the authors' analysis.

The new analyses described in response to comment (2) and new Figure 12 directly contradict the possibility that dynamic amplitude fluctuations of prominent (and largely sustained) beta oscillations that were wrongfully classified as epochs with high burst rate.

This appears particularly a concern given that most inter-burst-intervals (Figure 9) indeed appear to occur within the range of few millisecond only. Have the authors considered imposing a minimum interval between successive bursts (and/or a minimal amplitude relaxation between successive bursts)? Would the same results be obtained?

We thank the reviewer for the close reading, and confirm the reviewer’s observation that in our original definition of events, multiple events could occur within one continuous high-power (above power cutoff) region (see Figure 2). Per the reviewer’s suggestion, we considered an alternative definition of an event and applied the same analyses. In the new definition, we considered each continuous supra-cutoff temporal region in the spectrogram to be an event. With this approach, each successive event would be separated by sub-cutoff periods, effectively imposing a “minimal amplitude relaxation between successive bursts” as suggested by the reviewer. We term events using this redefinition “non-overlapping events.” As before, event power is defined as the maximal power within the supra-cutoff region, and event duration and frequency span are defined as the full-width at half-maximum in the time and frequency domain, respectively. Summary of our main results for non-overlapping events is plotted in new Figure 11, shown below.

Despite the complete abolition of the Inter-Event-Intervals (IEI) that “occur within the range of a few millisecond only” (Figure 11ii), we see in Figure 11 that most (~80%) of the “non-overlapping events” have only one local maxima in a single continuous supra-cutoff region. In other words, ~80% of “non-overlapping events” were also identified as an event in our original definition. Given this high congruency between the original event definition and the alternative “non-overlapping event” definition, it is not surprising that we found most of the results obtained from the original definition of events to hold in this redefinition. The following is a further elaboration of the agreement in results.

First, we find that the “non-overlapping events” do not occur rhythmically. The IEI histogram starts at zero and rises to a peak near 200ms (Figure 11ii), likely due to the refractory period imposed by the constraint of “minimal amplitude relaxation between successive bursts”. Alternatively, it is possible that the peak near 200ms was reflective of underlying rhythmicity. To test for the presence or lack of rhythmicity in IEI of “non-overlapping events”, we constructed jittered IEI distributions (Figure 11ii, black curve, see Figure 11 legend for details). Neither the IEI distributions in the detected nor the non-detected (the attend-in nor the attend-out) trials are distinct from the jittered distributions, suggesting that non-overlapping events do not occur rhythmically in either behavioral condition. This further corroborates our original conclusion that beta event generation in our data is not rhythmic. Note, these conclusions are again limited by our restriction to the 1 second prestimulus period and the fact that only trials with ≥ 2 events were considered (i.e., ~30% of the trials, Figure 5), as was the case for the original definition of events.

We further tested whether the conclusions regarding the influence of beta events on mean power and behavior held with this redefinition. In good agreement with our original definition of events, the number of “non-overlapping events” best correlated with trial mean prestimulus beta power (Figure 11iii, Friedman test followed by Wilcoxon signed rank test, with Holm-Bonferroni correction) and was the most consistent predictor of behavior across species and tasks, as measured with detect / attend probabilities (Figure 11iv).

We have added a new Figure 11 and describe it in the “BETA events did not occur rhythmically” section of Results, and Materials and methods.

Figure 5 and Figure 9 (duration ~ 100 ms and intervals of <50 ms) together suggest that many bursts occur as doublets of about 250 ms, or triplets. Is that so? If yes, there should be more to characterize, like the timing of doublets. It may be noted that 250 ms or less is about the duration of the prestimulus period described as when beta events are effective in modulating sensory detection.

We think there was some confusion about our definition of IEI. It appears the reviewer interpreted the IEI to be the interval between the end of a previous burst and the beginning of the next burst (hence the 100+50+100=250ms calculation). Rather, our definition was based on timing between two consecutive maxima in the beta band (see Materials and methods). As such, two or more maxima (beta events) could occur within one continuous supra-cutoff region, and create an IEI of <50ms. Per this comment, and that above, we performed a new analysis of IEI with “non-overlapping events” (see above). This analysis revealed that most supra-cutoff regions had a single event (~80%). In particular, this high preservation of events from the old to the new definition was non-differential for behavioral outcomes / conditions; i.e. given an event, the probability that it would be a multi-maxima event was not higher or lower in non-detected trials (Figure 11). Therefore, we would not expect the single-maxima events (which is an event in both the old and the new definition) and the multi-maxima events (which is multiple events in the old definition but one event in the new definition) to influence behavior differentially.

In addition, Figure 9 shows the interval distributions do not differ between behavioral outcomes. It looks inconsistent with that more beta events occur in miss trials than hit trials.

We thank the reviewer for pointing out that one would typically expect rate and inter event interval (IEI) distribution to be related. The apparent discrepancy stems from the constraint that IEI by definition could only be calculated in trials that had ≥2 events in the 1 second prestimulus window, and most (~70%) trials had 0 or 1 events (Figure 5). As such, the IEI analysis was restricted to only the remaining ~30% trials with ≥2 events, and we cannot expect the typical rate to interval relation to hold.

4) For comparison, it appears key to also include outcomes of regular power analyses when relating neural activity to behavioral performance, as in Figure 7 (for the 1s pre-stimulus epoch) and Figure 8 (power as a function of time). Are events features (rate) really better predictors of performance than conventional power estimates?

We thank the reviewer for this comment and would like to first emphasize that the goal of our analysis was to determine which event feature(s) contribute to statistically significant differences in average prestimulus power, rather than to determine if any one of the features was a “better” predictor than averaged power. Many prior studies, including our own, have established that averaged prestimulus power robustly predicts behavior. Yet, knowledge of how beta rhythms impacts behavior remains relatively unknown. In our view, a key step in uncovering beta’s role in function is to understand features of the prestimulus ‘rhythm’ that contribute to observed average power differences.

We note that Figure 1 in our paper is already one of the figures requested by the reviewer: “outcomes of regular power analyses when relating neural activity to behavioral performance, as in Figure 7 (for the 1s pre-stimulus epoch)”.Comparing Figure 1 results to Figure 7 results shows event number is not a better (nor worse) predictor of perception / attention than averaged power. When considering averaged prestimulus power, 50% of subjects in human detection and 50% sessions in mouse detection had DP significantly less than 0.5, and 10% of subjects in human attention had AP significantly less than 0.5; in comparison, the% of subjects / sessions values for event number were 40%, 40%, and 20% in each dataset, respectively (see Figure 1iv parentheses).

In addition, a newly added analysis calculating the effect size as Cohen’s d (see our response to reviewers’ comment (6), and Materials and methods) shows that effect sizes are similar for trial mean prestimulus beta power (1.3 for human detection, 1.69 for mouse detection, 0.94 for human attention; see Figure 1iii) and event number (1.15 for human detection, 1.97 for mouse detection, 0.88 for human attention; see Figure 7iii).

We agree with the reviewer that it is also important to display averaged power as a function of time, to show that the 1 second prestimulus window was an appropriate choice for assessing beta’s impact on function, and to assess whether beta power difference is more prominent closer to the time of the stimulus. We have added this additional analysis as a new panel in Figure 1 (Figure 1), which shows that beta power is higher on non-detected / attend-out trials throughout the prestimulus window. Further, new Figure 7 depicts event rate time histograms across the 1 second prestimulus window, which similarly supports that differences in the rate of events across behavioral outcomes / conditions are distributed in the prestimulus period.

The results pertaining to new Figure 1 and Figure 7 are discussed in the Results section.

It is worth noting that these added figures are distinct from Figure 8, which depicts the histogram of the timing of the single beta event closest to the stimulus (i.e., a single “most recent event” per trial), rather than a time histogram of the event rate of all events before the stimulus (Figure 7).

5) The authors limit analysis to the beta frequency range, but in the discussion mention alpha and gamma bands. One might expect these frequency ranges to be modulated as well at least in a fraction of sensors and recording sites. Why did the authors not add a brief summary analysis on whether similar findings as the beta event rate changes might be obtained with alpha and gamma ? In that way one immediately can discern whether the findings have clear frequency specificity, or whether similar effects can be expected in other bands with clear power spectral peaks. Either of these results would be highly informative.If the authors find this analysis would exceed the scope of the study, the discussion on alpha and gamma should be shortened and instead a discussion of the possibly different functional roles of the beta frequency (sub)ranges is included to discern better why the analysis concentrates on this band and whether the authors suggest that the band represents some homogeneous underlying sources (or not).Also, there is evidence that rolandic alpha oscillations also predict somatosensory performance and are modulated by somatosensory attention. Have the authors considered extending their analyses to this frequency range?

We agree with the reviewer that an investigation of other frequency bands is of high interest. ALPHA in particular is relevant in our data, as our prior publications show that averaged SI alpha power also shifts with attention and predicts perception (Jones et al., 2010). Further, alpha band activity also appears to be event like in our data (Jones et al., 2009), however this fact has not yet been fully characterized. Of note, in our macroscale human data gamma power is not distinguishable from the dominant 1/fn trend.

While of high interest, we feel a full analysis of alpha band activity is beyond the scope of our current study. First, our group has recently established a new theory about the origin of transient neocortical beta rhythms (Sherman et al., 2016). Therefore, as a direct next analysis, we were most interested in assessing the behavioral impact of such transient properties for beta rhythms. Second, we have observed that alpha and beta have separable effects on attention and perception. Shifts in averaged alpha power occur sooner after attentional cues than shifts in beta power, and beta showed a stronger linear relationship to detectability (Jones et al., 2010, Sacchet et al., 2015). We view these alpha dynamics as distinct and equally important as beta dynamics, and deserving of a full analysis. Our paper is already covering many new ideas and analysis methods, and includes beta dynamics across two tasks, two recording modalities and two species. As such, we think including alpha would expand beyond the scope of this study but provides food for future analysis.

To directly follow the reviewer’s advice, we have shortened the discussion of alpha and gamma oscillations and updated the text in the Discussion:

“Of note, our data also show high power alpha band activity (Jones et al., 2010, Jones et al., 2009). We have observed that alpha and beta have separable correlations with attention and perception. Shifts in averaged alpha power occur sooner after attentional cues than shifts in beta power, and beta showed a stronger linear relationship to detectability (Jones et al., 2010, Sacchet et al., 2015). A full quantification of the potential transient nature of alpha and its relationship to beta events would be an interesting future direction.”

We also agree with the reviewer that the potential import of ‘(sub) ranges of beta’ merits a more focused discussion than was present in the original paper. In our data, we consistently find only a single distinct ‘bump’ in the power spectra in the beta band, as shown in prior publications from our group (Jones et al., 2009). We also believe that SI beta that modulates somatosensory perception is generated by a unique mechanism that manifests as homogeneous, stereotyped waveform features (Sherman et al., 2016). To directly meet the concerns of the reviewer, we now discuss this homogeneity in the beta band as follows:

“In our human data, power-spectral density analysis shows a single “bump” in the alpha band (7-14 Hz) and a single bump in the beta band (15-29 Hz); whereas the gamma (30-80 Hz) power is not distinquishable from the1/fn trend (Jones et al., 2009). The bump in the beta band emerges primarily between 18-22 Hz, also known as the beta 2 range, and we do not observe a separate bump for the beta 1 range (typically ~15Hz; (Kopell et al., 2014)). We conjecture that this isolated increase in the power spectral density in the beta band is indicative of a singular underlying mechanism. In further support, as discussed above, we have previously shown evidence for a unique mechanism that generates stereotyped and homogeneous beta-band activity (Sherman et al., 2016).”

6) What was the proportion of trials in each dataset with at least one event (the inclusion criteria) in correct and error trials? Without this information, it is difficult to discern how important the beta events would be to predict trial-by-trial performance. A discussion of the overall effect size considering this number would be very interesting to readers – this is particularly relevant with regard to the stimulation approaches that would want to utilize event like stimulation protocols.

As shown in Figure 5, the proportion of trials with zero events was consistently around 0.4 (i.e. 40%) in each subject / session in all three datasets. To directly address the reviewer’s suggestion that we report the proportion separately on detected and non-detected (attend-in and attend-out) trials, we have added a new Figure 7—figure supplement 2.

We agree with the reviewer that in thinking about stimulation approaches that would want to utilize event like stimulation protocols, it would be useful to think about the optimal event number criterion for classifying behavioral conditions. We performed this analysis and added the results as a new Figure 7—figure supplement 2 (see legend for definition of criterion). The optimal criterion where the classifier achieved best performance was 2 events for most subjects / sessions; 60% of subjects for human detection, 60% of sessions for mouse detection, and 50% of subjects for the human attention dataset had 2 as the optimal event number criterion.

We also added calculation of the effect size, which quantifies the difference between behavioral conditions; i.e. (miss – hit) for detection datasets and (out – in) for attention dataset. We employed the Cohen’s d measure, which is defined as follows:d=x´1−x´2(n1−1)s12+(n2−1)s22n1+n2−2

where x´1 and x´2 denote the mean of the two populations (e.g. miss vs hit) being compared, an n1 and n2 denotes the number of sessions in each population, and s1 and s2 are the standard deviation of each population. (Materials and methods).

Per reviewer’s suggestion, we calculated the effect size of the difference between behavioral conditions for the proportion of trials with 1 or more events. Given that our optimal criterion was 2 for most subjects / sessions, we also calculated the effect size for the proportion of trials with 2 or more events (new Figure 7—figure supplement 2). Although the effect size for the proportion of trials with 1 or more events was quite high, the proportion of trials with 2 or more events had a higher effect size in all three datasets.

It is worth noting in Figure 7—figure supplement 2 that the proportion of trials with 1 event was not substantially different between behavioral outcomes / conditions. Likewise, the corresponding effect size for the proportion of trials with 1 event was not high (new Figure 7—figure supplement 2). In contrast, the effect size for the proportion of trials with 0 events (which is same in absolute value but negative in sign of effect size for the proportion of trials with 1 or more events) was much larger in absolute value. Therefore, we surmise that stimulation protocols that drive two or more events in the 1 second period may impair perception, whereas blocking events from happening in the 1 second prestimulus period (i.e. 0 events) may benefit perception.

In trials where there is just one event (~30% of all trials, Figure 5), other event features may contain more information about the behavioral condition. In light of Figure 8, we would expect the timing of the event to contain further information about the likelihood of detection. Indeed, new Figure 7—figure supplement 2 shows that the event timing is significantly (p<0.05) closer to the stimulus onset on non-detected trials in the human detection dataset, and trends toward significance (p<0.1) in the mouse detection dataset (right-tailed paired t-test). We would not expect there to be a difference in timing relative to the stimulus onset in the different attention conditions, as the cue instructing where to attend (attend-in vs attend-out) was ultimately what lead to the differences in beta dynamics, and the timing of the cue relative to the stimulus onset was randomized in this task (Figure 7—figure supplement 2).

Taken together, we conclude that the actual number of events contains more information than quantification of the mere presence or absence of events.

These new results are now described in Results section “Non-detected trials were more likely to have a prestimulus beta event closer to the time of the stimulus*”*, and discussed in the Discussion section.

Do the curves in Figure 8 derived from all trials or a fraction of trials that had beta events? Event rate can be derived in both ways. Blue and red curves in the figure should be derived from all detected and non-detected trials respectively.Which trials were used to derive the threshold in Figure 4? All Hit, Miss, FA, and catch trials?

We apologize that the explanation of Figure 8 was not clear. We did indeed derive the blue and red curves from all detected and non-detected trials, respectively. We have added this fact to the Figure 8 legend and the following clarifications to the Materials and methods:

“…With the exception of Figure 8, only the trials that had at least one event was considered for all analyses pertaining to the “most recent event”.”

“Figure 8 depicts the probability of the “most recent event” happening at a certain time bin, given a detected / non-detected trial. Therefore, we included all detected / non-detected trials (at perceptual threshold) for this analysis.”

Which trials were used to derive the threshold in Figure 4? All Hit, Miss, FA, and catch trials?

We thank the reviewer for pointing out the omission of this important piece of information. All analyses were limited to trials at perceptual threshold, with the exception of new Figure 1—figure supplement 1. This applied to Figure 4 as well. We have clarified this in the “Mean prestimulus beta power is higher on non-detected and attend-out trials” section of Results:

“We restricted all further analyses to trials at perceptual threshold.”

Would the strength of the correlations look very different when including these zero event trials as zero's and using Spearman instead of Pearson correlations?

The reviewer is correct to point out that the comparison between event number and other parameters (trial mean event power, duration and frequency span) might be biased by the fact that different proportion of trials were used; all trials were analyzed for event number, whereas only the trials with at least one event was used for the other parameters. To directly address this concern we performed two additional analyses, as described further below: (1) the Spearman analysis as suggested, and (2) a multiple linear regression analysis. These further analyses corroborated our original findings (Author response image 2 and Author response image 3). However, we decided to keep the original analyses (Pearson’s correlation with zero event trials excluded for trial mean event power, duration and F-span) in the paper, as setting these values as zero in the large number of zero event trials (40% see Figure 5) would significantly underestimate their average values and ultimately not change the conclusions of our study, as elaborated below.

We have however made two adjustments in the paper to strengthen our statistical analyses pertaining to Figure 6. First, we employed the Friedman’s test before applying the pairwise Wilcoxon signed rank tests. We chose the non-parametric Friedman’s test rather than repeated measures ANOVA (rmANOVA) since rmANOVA assumptions of normality and equal variance were not consistently met. Second, we have corrected for multiple comparisons in the pairwise Wilcoxon signed rank tests. Further details are described in our response to reviewers’ comment (13) and in the revised paper (Results section and Materials and methods).

1) Here, we detail our application of the Spearman’s correlation as suggested by the reviewer (Author response image 2). We first calculated the Spearman’s correlation with trial mean prestimulus beta power, for each subject / session in each dataset. All trials were included for all features by setting trial mean event power, duration and frequency span as zero for zero event trials.

Before assessing statistical differences between distributions of Spearman’s correlation values with rmANOVA, we checked for rmANOVA assumptions; i.e. the normality assumption and the equal variance assumption. We checked for normality using the Lilliefors test. Of the 12 (4 features, 3 datasets) distributions of Spearman’s correlation values, all but the trial mean event duration in the human detection data (p=0.026) were not significantly different from a normal distribution. On the other hand, Bartlett’s test revealed that the equal variance assumption was valid for human detection (p=0.578) and mouse detection (p=0.468) datasets, but not for human attention dataset (p=8.52×10−11). Subsequent rmANOVA revealed significant (p<0.05) difference between event parameters in their correlation with trial mean prestimulus beta power (p=2.59×10−10,df=3,F=42.1 for human detection dataset; p=1.14×10−15,df=3,F=118.8 for mouse detection dataset; p=8.52×10−11,df=3,F=46.5 for human attention dataset).

Since the normality assumption was violated for the Spearman’s correlation value distributions of trial mean event duration in the human detection dataset, and the assumption of equal variance was violated for the human attention dataset, we deemed it more appropriate to conduct the Friedman test rather than the rmANOVA test. The Friedman test also revealed significant correlations in all three datasets (p=9.36×10−6,df=3,χ2=26 for human detection dataset; p=5.89×10−6,df=3,χ2=27 for mouse detection dataset; p=9.36×10−6,df=3,χ2=26 for human attention dataset).

To compare across features as in the original Figure 6, we conducted post-hoc pairwise comparisons using Wilcoxon signed-rank test, with Holm-Bonferroni corrections. Significance determined at p<0.05 are summarized with asterisks in Author response image 2. Results are in general agreement with our original analysis, with the difference that event number and event power comparison was not significant in the human detection dataset nor the human attention dataset.

**Author response image 2. respfig2:** Spearman’s correlation with trial mean prestimulus beta power. All trials were included for all features, and trial mean event power, duration and frequency span as zero for zero event trials.

2) For thoroughness, we also conducted multiple linear regression with event number, trial mean event power, duration and F-span as regressors and the trial mean prestimulus beta power as the dependent variable. Multiple linear regression is appropriate here since all trials are included for all event parameters, and it has an advantage over Spearman’s correlation in that it can assess the relative contribution of each regressor while controlling for other regressors. To keep the spirit of Spearman’s (rank) correlation, the multiple linear regression was done with trial ranks as regressors, where the trial ranks were based on each event feature respectively, rather than raw values for trial summary of event features. The following Author response image 3 depicts the slope values for each regressor, where multiple linear regression was done separately for each subject in each dataset.

To assess significance, here we applied rmANOVA since assumptions of normality and equal variance were found to be valid in all cases, and found that the regression slope distributions were significantly different in all three datasets (p=1.43×10^-14^, df=3, F=96.9 for human detection dataset; p=8.03×10^-20^, df=3, F=250.9 for mouse detection dataset; p=8.49×10^-18^, df=3, F=174.9 for human attention dataset). To compare across features, we applied post-hoc paired t-test with Holm-Bonferroni corrections; since the normality assumption was valid, we applied this parametric test rather than Wilcoxon signed-rank test. Results are depicted with asterisks, p<0.05 in Author response image 3. In agreement with our original analysis (Figure 6), event number rank was a significantly better regressor of trial mean power rank in terms of slope compared to ranks of trial mean event power, duration and F-span in all three datasets. This suggests that the superiority of event number in its correlation with mean power is not solely driven by the fact that it had a wider dynamic range than other features it was being compared to.

**Author response image 3. respfig3:** Multiple linear regression slopes, with trial mean prestimulus beta power as the response variable and the four trial summary event features as regressors. All trials were included for all four features; trial mean event power, duration and frequency span were defined as zero for zero event trials.